



# Direct contribution of ammonia to CCN-size αlpha-pinene secondary organic aerosol formation

Liqing Hao[1], Eetu Kari[1,a], Ari Leskinen[1,2], Douglas R. Worsnop[1,3], Annele Virtanen[1]

[1]Department of Applied Physics, University of Eastern Finland, Kuopio, Finland
[2]Finnish Meteorological Institute, Kuopio, Finland
[3]Aerodyne Research Inc., Billerica, MA 08121-3976, USA
[a] now at: Neste Oyj, Porvoo, Finland

*Correspondence to*: Liqing Hao (hao.liqing@uef.fi) and Annele Virtanen (annele.virtanen@uef.fi)

**Abstract.** Ammonia (NH₃), a gasous compound ubiquitously present in the atmosphere, is involved in the formation of secondary organic aerosol (SOA), but the exact mechanism is still not well known. This study presents the results of SOA experiments from the photooxidation of α-pinene in the presence of NH₃ in the reaction chamber. SOA was formed in nucleation experiment and in seeded experiment with ammonium sulfate particles as seeds. The chemical composition and time-series of compounds in the gas- and particle- phase were characterized by an on-line high-resolution time-of-flight proton transfer reaction mass spectrometer (HR-ToF-PTRMS) and a high-resolution time-of-flight aerosol mass spectrometer (HR-ToF-AMS), respectively. Our results show that for the aerosol particles in cloud condensation nuclei (CCN) size, the mass concentration of ammonium (NH₄⁺) was still rising even after the mass concentration of organic component started to decrease due to aerosol wall deposition and evaporation, implying the continuous new formation of particle phase ammonium in the process. Stoichiometric neutralization analysis of aerosol indicates that organic acids have a central role in the formation of particle phase ammonium. Our measurements show a good correlation between the gas phase organic mono- and di-carboxylic acids formed in the photooxidation of α-pinene and the ammonium in the particle phase, thus highlighting the contribution of gas-phase organic acids to the ammonium formation in the CCN-size SOA particles. The work shows that the gas-phase organic acids contribute to the SOA formation by forming ammonium salts through acid-base reaction. The changes in aerosol mass, particle size and chemical composition resulting from the NH₃-SOA interaction can potentially alter the aerosol direct and indirect forcing and therefore alter its impact on climate change.

## 1 Introduction

The largest uncertainty in forward projection of global warming is related to our limited knowledge of negative solar radiative forcing associated with aerosols (IPCC, 2013). Formation of secondary organic aerosols (SOA) is one of the main processes that affects the composition and properties of atmospheric aerosols. Formation of SOA occurs through two distinct mechanisms: by increasing the mass of the existing aerosol and through the formation of new particles. Estimate on the SOA formation shows its significance as a source of atmospheric organic aerosol: about 60% of the organic aerosol mass is SOA on the global scale and regionally even more (Hallquist *et al*., 2009; Jimenez et al., 2009; Kanakidou *et al*., 2005). Hence, SOA plays an important role in the direct scattering of solar radiation, cloud formation and precipitation, and visibility reduction, and may also have a direct impact on human health.

Ammonia (NH₃) is ubiquitously present in the atmosphere as a dominant volatile base. The majority of its sources is accounted by emissions from agriculture, livestock, soil and traffic (Huang et al., 2012; Grönroos et al., 2009; Battye et al., 2003). NH₃ governs the neutralization of atmospheric aerosol by reacting with the inorganic acids such as sulfuric acid and nitric acid, leading to transformation of a substantial amount of ammonium sulfate (and



derivatives) and ammonium nitrate (Seinfeld and Pandis, 2016). These inorganic salts play a vital role in contributing
to the fine particle matter (PM2.5) and altering the chemical and physical properties of aerosol particles in the
atmosphere. The mechanism between $NH_3$ and the inorganic acids leading to the secondary inorganic aerosol has
been well recognized.

A number of studies have shown that $NH_3$ is one of the key species for the new particle formation through ternary
and binary nucleation with water and sulfuric acid (e.g. Lehtipalo et al., 2018; Jokinen et al., 2018; Bianchi et al.,
2016; Kirkby et al., 2011; Kurten et al., 2007; Kulmala et al., 2000). The nucleated particles are a significant source
of atmospheric SOA particles and subsequent growth to a larger size (>50nm) allows them to serve as cloud
condensation nuclei (CCN). However, the role of $NH_3$ for the CCN-size SOA particles are still rarely studied. A study,
conducted more than a decade ago, demonstrated that $NH_3$ increased the number and volume concentrations of CCN-
sized SOA particles from α-pinene-ozone system by 15% and 8%, respectively, (Na et al., 2007). Similar results have
also been observed in the photooxidation and ozonolysis of a-pinene SOA experiments that SOA mass yield increased
by 13% as a response to $NH_3$ addition (Babar et al., 2017). Besides, the addition of $NH_3$ could also promote the SOA
formation from photooxidation of vehicle exhaust (Chen et al., 2019; Liu et al., 2015), anthropogenic VOCs (Wang
et al., 2018; Huang et al., 2018) and acrolein (Li et al., 2019). The promotion mechanism of $NH_3$ to SOA formation
can be both through base-acid reaction (Schlag et al., 2017; Babar et al., 2017; Na et al., 2007) and by the $NH_3$ uptake
to the carbonyl group (Zhu et al., 2018; Liu et al., 2015). As a consequence, the changes in particle size and chemical
composition could alter the CCN ability and hygroscopicity of SOA particles (Dinar et al., 2008). Moreover, the
reaction of $NH_3$ with SOA decreases the volatility of SOA particle (Paciga et al., 2014), and also results in production
of light-absorbing brown carbon compounds that modify the optical properties of the aerosols (Huang et al., 2018;
Updyke et al., 2012; Bones et al., 2010). Additionally, the updake of $NH_3$ by SOA can deplete ambient $NH_3$
concentrations, causing indirect reductions in the amount of inorganic ammonium salts in particulate matter (Horne
et al., 2018). Therefore, the interaction of $NH_3$ and SOA could alter both direct and indirect aerosol radiative forcing
and potentially alter its impact on climate change.

This work presents the results of SOA formation from photooxidation of α-pinene in the presence of $NH_3$ in the
nucleation and seeded experiments. The chemical composition of gas-phase and particle-phase compounds were
characterized with a high-resolution time-of-flight proton transfer reaction mass spectrometer (HR-ToF-PTRMS) and
a high-resolution time-of-flight aerosol mass spectrometer (HR-ToF-AMS), respectively. Our experiments show the
formation of ammonium salt ($NH_4^+$) in the CCN-size SOA particles, and our gas phase measurements indicate organic
acids are responsible for their formation. The results will have potential applications in studying SOA formation
mechanism and the impact of SOA on climate forcing.

## 2    Experimental

### 2.1 Reaction chamber experiments

SOA experiments were conducted in the ILMARI environmental chamber infrastructure in University of Eastern
Finland. The experimental system and experimental procedure have been described in details in Leskinen et al. (2015)
and Kari et al. (2019a), respectively. The chamber consists of a 29 $m^3$ Teflon$^{TM}$ FEP film bag. Ultraviolent (UV)
lights with a spectrum centered at 340 nm around the chamber enable photochemical reactions. Two sets of
experiments were designed (Table 1): (1) SOA nucleation experiments in the absence of seed aerosols; (2) SOA
formation experiments in the presence of ammonium sulfate seeds. Prior to each experiment, the chamber was





continuously flushed overnight with laboratory clean air produced by a zero air generator (Model 737-15, Aadco Instruments Inc., USA) and conditioned in a humidifier (Model FC125-240-5MP-02, Perma Pure LLC., USA),

aiming at 50 % relative humidity at the typical temperature of 20 °C during the course of each experiment. Independent on the type of experiments (nucleation and seeded), the reactants were added into the chamber in the sequence described below. The $(NH_4)_2SO_4$ (ammonium sulfate, 99 %, Sigma Aldrich) seed was injected to the chamber first. The seed was generated using an atomizer (Topas ATM226, Germany). In two experiments, nitrogen oxide (NO) was added to chamber, and ozone ($O_3$) was introduced into the chamber to convert NO to nitrogen dioxide

($NO_2$) to reach the atmospherically relevant $NO_2$-to-NO ratio of ~ 3 (Kari et al., 2019a). After $O_3$ and $NO_x$ were fed, 3 µl 9-fold butanol (butanol-d9, 98 %, Sigma Aldrich) was injected to the chamber, from whose consumption the hydroxyl radical (OH) exposure was estimated in each experiment (Kari et al., 2019a). Next, 1 µl and 18 µl of α-pinene (≥ 99%, Sigma Aldrich) were added into the chamber for the seeded and nucleation SOA experiments, respectively, corresponding to concentrations of ~5 ppb and ~100 ppb of α-pinene in the chamber. Last, 5 ml of

hydrogen peroxide ($H_2O_2$, 30 wt. % in $H_2O$, Sigma Aldrich) was conducted into the chamber by mixing it with purified air flowing at 10 lpm (liter per minute), as a precursor for OH radicals to be generated under UV radiation. After all the compounds were introduced into the chamber, the chamber was closed, and the compounds were allowed to stabilize for 15 min. Then the UV lights were switched on to initiate photochemistry.

### 2.2 Analytical methods and instrumentation

The size-resolved chemical composition and mass concentration of aerosol particles were measured directly with an on-line high-resolution time-of-flight aerosol mass spectrometer (HR-ToF-AMS, abbreviated as AMS) (DeCarlo et al., 2006). A detailed description of AMS operational procedure is provided in previous publications (Canagaratna et al., 2007; Jayne et al., 2000). In brief, AMS was operated in V - mode in EI - mode. Calibration of ionization efficiency (IE) followed the standard protocol using dried and size-selected ammonium nitrate particles. The data

were analyzed using standard AMS data analysis toolkits (Squirrel V1.62D and Pika V1.22D) in Igor Pro Software (version 6.37, WaveMetrics Inc.). For determining mass concentrations, the default relative ionization efficiency (RIE) values were 1.4, 1.1 and 1.2, for organics, nitrate and sulfate, respectively. The RIE for ammonium was 2.95, as determined in the IE calibration. After a comparison to the volume concentration derived from a scanning mobility particle sizer (SMPS TSI 3081 DMA + 3775 CPC) measurement (Fig. S1), a collection factor of 100 % was applied

to determine the aerosol mass concentration in the reported results in this work. The positive matrix factorization (PMF) analysis was performed on the high-resolution mass spectra by using the PMF Evaluation Tool V2.08 (Paatero and Tapper, 1994; Ulbrich et al., 2009). The standard error matrices were processed following the principles of applying minimum error estimate, downweighting weak variables, removing bad variables, and downweighing m/z 44 related fragments (Ulbrich et al., 2009). The PMF was evaluated with 1 to 6 factor. Rotation (Fpeak) varied from

-1 to 1 at a step of 0.1.

The concentrations of α-pinene and oxidized organic products were quantified by an on-line high-resolution time-of-flight proton transfer reaction mass spectrometer (HR-ToF-PTRMS, abbreviated as PTRMS) (Ionicon Analytik). A detailed description of the instrument and operation procedure has been provided in Jordan et al. (2009) and Kari et al. (2019a; 2019b). In brief, the PTRMS instrumental setting were the same as in Kari et al. (2019a). During

measurement, the mass calibration was conducted using protonated water isotope signals at m/z 21 and internal instrumental signals of diiodobenzene and its fragment ions (protonated integer m/z 331). The instrumental



transmission calibration was conducted using a standard gas mixture containing 8 different VOCs (protonated integers ranging from m/z 79 to m/z 181). The concentration of VOCs were determined according to the principles by Hansel et al. (1995). The selected ions that we are interested in this study are consistent with α-pinene (m/z 136+1), formic acid (m/z 46+1), acetic acid (m/z 60+1), propionic acid (m/z 74+1), pinonic acid (m/z 184+1), butyric acid (m/z 88+1), pentanoic acid (m/z 102+1), malonic acid(m/z 104+1) and succinic acid (m/z 118+1).

For supporting information we measured the particle concentration and size distribution in a diameter range of 7-800 nm with an SMPS (TSI 3081 DMA + 3775 CPC) and the concentrations of NO, nitrogen oxides (NO$_x$), ozone O$_3$, and sulfur dioxide (SO$_2$) , as well as the relative humidity and temperature inside the chamber. In this study, we lacked the measurement of NH$_3$ concentration in the chamber but estimated it from our AMS measurement results, by assuming that the particulate ammonium salt (NH$_4^+$) was converted from the gas-phase NH$_3$ (Fig. 1). The maximum NH$_4^+$ concentration was in the range of 1.17~1.51 μg m$^{-3}$, which corresponds to a minimum NH$_3$ concentration level of 1.6 ~ 2.1 ppbV in our chamber.

## 3 Results and discussion

### 3.1 Time series of aerosol species

Aftert the UV lamps were switched on, the photooxidation reaction produced oxidized gas-phase compounds and SOA particles in both nucleation and seeded experiments. The time series of mass concentrations of formed SOA (in green) and ammonium (in orange) along with sulfate and nitrate components measured by AMS are presented in Fig. 1. In the nucleation experiments (the left panels in Fig. 1), we observed rapid increase in the SOA mass concentrations after the photooxidation reaction had started. . After reaching the maximum concentration of 390 - 476 μg m$^{-3}$, the SOA concentration declined by 27.9 ± 9.2 % at the end of experiment because of particle deposition on the chamber wall and/or aerosol evaporation. The O:C ratio (oxygen to carbon ratio) of the SOA particles were slightly increased from the initial 0.39 ± 0.015 to the final 0.44 ± 0.01 because of aerosol aging. In a distinct contrast, the mass concentrations of ammonium component were still rising at the stage of decreasing SOA masses. Together taking into account the fact that aerosol wall deposition loss was present resulting in decreasing organic mass (and decreasing sulphate mass in the seeded experiments), our results suggest new production of ammonium salts. The newly formed ammonium can be partly attributed to the co-generated nitrate and sulfate as the photooxidized products of NO and SO$_2$ in the chamber. However, the amount of the two inorganic species can't fully interpret the ammonium and we will elaborate this in more detail in Sec. 2. Similar phenomena were also observed in the seeded experiments (Fig. 1 and Fig. 2). We need to point out that the lower cutoff diameter of SOA particle is about 35nm for AMS measurement (Zhang et al., 2004), and the majority of the SOA mass is dominated by particles in the Aitken- and accumulation modes. Thus the reported results are with SOA in CCN size in this study.

### 3.2 Participation of ammoina in SOA formation

### 3.2.1 Reaction of ammoina with organic acids

To investigate the monotically increasing profile of the ammonium salt in our chamber experiment, we studied the stoichiometric neutralization of formed SOA particles. The approach proposed by Zhang et al (2007) was adopted, in which the ammonium mass concentrations measured (NH$_4^+$$_{,mea}$) in the particles were compared to the stoichiometric ammonium concentrations required (NH$_4^+$$_{,pre}$) to fully neutralize the measured concentrations of SO$_4^{2-}$, NO$_3^-$ and Cl$^-$:





$\quad$ $NH_4^+{}_{,pre} = 18\times(2\times SO_4^{2-}/96+NO_3^-/62+Cl^-/35.5)$ $\qquad$ (1)

where $NH_4^+$, $SO_4^{2-}$, $NO_3^-$ and $Cl^-$ represent the mass concentrations (in µg m⁻³) of the species and the denominators correspond to their molecular weights. The factor 18 is the molecular weight of $NH_4^+$.

A comparison between the predicted and measured ammonium masses is displayed in Figs. 2 and S2. In both sets of experiments, the measured ammonium mass concentration was systematically greater than the predicted value.

$\quad$ The trend doesn't show a dependence on the presence of $NO_x$ in the chamber. On average, $NH_4^+{}_{,mea}$ is $400 \pm 156$ % and $21 \pm 11$ % greater than $NH_4^+{}_{,pre}$ at the end of nucleation and seeded experiments, respectively. The large discrepancy between the measured and predicted ammonium concentrations suggested that the current amount of sulfate, nitrate and chloride is insufficient to neutralize the ammonium formed in the particle phase, which indicates that organic component must have played a role in this process. Considering the nature of $NH_3$ as a base compound,

$\quad$ the candidate species of organic compounds are attributed to organic acids.

Since a vast variety of molecular compositions of organic acids may be present in the photooxidation products of a-pinene, it isn't possible to define the amount of individual organic acid required to neutralize the $NH_3$. Therefore, we use the $CO_2^+$ ion measured by AMS to represent carboxylic functional group of organic acids. The $CO_2^+$ is not only considered as a reliable marker of oxygenated organic aerosol (e.g. Zhang et al., 2005), but is also tightly

$\quad$ associated with the formation of organic mono- and di-acids shown in laboratory and field measurements (Yatavelli et al., 2015; Takegawa et al., 2007; Alfarra et al., 2004).

Taking into account the contribution of organic acids to ammonium salt, we reformulate Eq. (1) to:

$NH_4^+{}_{,pre} = 18\times(2\times SO_4^{2-}/96+NO_3^-/62+Cl^-/35.5+CO_2^+{}_{\_NH4}/44)$ $\qquad$ (2)

where $CO_2^+{}_{\_NH4}$ is the mass concentration of carboxylic function group (-$CO_2$) representing organic acids which were

$\quad$ required to neutralize the ammonium. The denominator 44 is the molecular weight of carboxylic functional group.

To interpret $NH_4^+{}_{,mea}$, $NH_4^+{}_{,pre}$ should equal to $NH_4^+{}_{,mea}$:

$NH_4^+{}_{,pre} = NH_4^+{}_{,mea}$ $\qquad$ (3)

We can estimate the amount of $CO_2^+{}_{\_NH4}$ by combining Eqs. (2-3):

$NH_4^+{}_{,mea} = 18\times(2\times SO_4^{2-}/96+NO_3^-/62+Cl^-/35.5+CO_{2\_NH4}^+/44)$ $\qquad$ (4)

$\quad$ So, $CO_2^+{}_{\_NH4}=(NH_4^+{}_{,mea}/18- 2\times SO_4^{2-}/96 -NO_3^-/62-Cl^-/35.5) \times44$ $\qquad$ (5)

The time series of estimated $CO_2^+{}_{\_NH4}$ over each experiments is shown in Fig. 3. On average, the $CO_2^+{}_{\_NH4}$ concentration required to explain the observed ammonium concentrations was 48.6 times lower for seeded experiments than for the nucleation experiments. Fig. 3 indicates that organic acids participated in reacting with $NH_3$ much earlier in the low-$NO_x$ test (red and green mark) than in the high-$NO_x$ experiment (blue mark). Based on the

$\quad$ time series in Fig. 3, in the high-$NO_x$ test, the time at which the organic acids started to play a role in ammonium formation was delayed by 31 and 100 minutes for the nucleated and seeded experiments, respectively. This observation can be associated to the formation of nitric acid ($HNO_3$) from photooxidation of $NO_x$ compounds in the high-NOx conditions. The reaction of $HNO_3$ and $NH_3$ takes precedence over the reaction between organic acids and $NH_3$. In general, the required $CO_2^+{}_{\_NH4}$ accounted for the $27.0 \pm 3.1$ % of total $CO_2^+$ mass in the nucleation

$\quad$ experiments and $18.7 \pm 6.0$ % in the seeded SOA experiments.

To further verify our conclusion that organic acids are the drivers of the ammonium formation, we explored the size distribution of organic acids (represented by $CO_2^+$ ion), ammonium and nitrate at the end of nucleation experiments (Fig. 4). The mode diameters of the three species, determined by performing log-normal fitting on the size distributions, are listed in Table S1. The mode diameter of $CO_2^+$ ion is about 5-13 nm greater than that of

$\quad$ ammonium and nitrate in the three individual experiments. The slight difference in two species mode diameters might





be associated with the lower evaporation rate of organic $CO_2^+$ than ammonium ions on the AMS vaporizer. Anyhow, the similarity in the mode diameters and size distributions of three chemical species suggests that they are internally mixed in the physical phase and are originated from the similar formation sources.

### 3.2.2 Connecting gas compounds by PTRMS to ammonium ion by AMS

To investigate the organic acid species which may potentially contribute to the ammonium formation in our experiments, we first examined the organic monocarboxylic acids in the gas phase formed in the photooxidaiton of α-pinene. Fig. 5(a1) shows the correlation between the concentration of butyric acid ($C_4H_8O_2$) (or its isomer) measured by PTRMS and the particle-phase ammonium salts measured by AMS in the nucleated SOA experiments. The excellent linear correlation (coefficient $R^2 \approx 1$) of these two species implies that the formation of butyric acid is

associated with the ammonium formation. In addition, we also identified a molecular ion $C_4H_8O_2^+$ in the AMS mass spectrum, which also shows an excellent correlation to the formed ammonium (Fig. 5c1) and thus indicates its simultaneous formation with ammonium ion. The simultaneous observation of the same ion molecules in both gas and particle phases gives us confidence to speculate that this ion is derived from gas-phase butyric acid. The observation suggests a reaction between butyric acid and $NH_3$ which enables the production of ammonium butyrate

salts ($NH_4C_4H_7O_2$). The formation of the salts favors the condensation of butyric acid on the particle phase contributing to the observed ammonium ions. Although we can't overlook the fact that ammonium butyrate is severely fragmented inside AMS, it is believed that the detected $C_4H_8O_2^+$ ion signal is a residual of parental molecule. In the same way, we also explored pentanoic acid ($C_5H_{10}O_2$) (or its isomer) (Fig. 3b1 and 3d1) and obtained similar results as with butyric acid.

A similar comparison has also been made in the seeded SOA experiments (Fig. 5a2-d2). Because of the presence of ammonium sulfate seeds, the ammonium attributed to the organic acids reaction (defined as $NH_{4,orgacid}$) is estimated to be the difference between $NH_4^+{}_{,pre}$ and $NH_4^+{}_{,mea}$. The butyric and pentanoic acids measured by PTRMS were then compared to the calculated $NH_{4,orgacid}$ in the particle phase. We have observed a good correlation of butyric acid to the organic acid-driven ammonium ($NH_{4,orgacid}$) ($r^2 = 0.68$-$0.73$), and a moderate correlation of pentanonic acid to

$NH_{4,orgacid}$ ($r^2 = 0.22$-$0.63$). Compared with the nucleation experiments, the correlation relationship in the seeded SOA experiments is worse, mainly because the signals in both PTRMS and AMS measurements are weak in the seeded SOA experiments. Similar to the observation in the nucleated SOA experiments, the ion $C_4H_8O_2^+$ in AMS mass spectrum was also identified to correlate to $NH_{4,orgacid}$. The ion has the same molecular formula as butyric acid. The concentration of $C_5H_{10}O_2^+$ of the same molecular formula as pentanoic acid was in a concentration level of $10^{-3}$ ug m$^{-}$

$^3$ (Fig. 5d2). Such signal is at the same level as the background noise in AMS measurement, making it challenging to correlate with $NH_{4,orgacid}$. In general, the results in the seeded SOA experiments are consistent with those in the nucleated SOA experiments, confirming the role of butyric and pentanoic acids in the formation of ammonium salt.

We also extend our study to other four types of gas phase organic acids measured by the PTR-MS. Fig. 4 shows the relationship of the formic acid, acetic acid, propionic acid and pinonic acid measured in the gas phase with the

ammonium ion measured by AMS. Surprisingly, the gas phase formic acid, acetic acid and propionic acid were in in good agreement with particle phase ammonium concentration with excellent linear correlation coefficients $r^2 \geq 0.89$ in the nucleated SOA experiments and $r^2 \geq 0.69$ in the seeded SOA experiments. The good correlation of gas phase organic acids with the particle phase ammonium salt suggests that theses acids played a role in the formation of ammonium. Formic acid and acetic acid are the most abundant organic monoacids in the atmosphere (Chebbi and

Carlier, 1996), whose one significant source is from photooxidation of a-pinene and other alkenes and terpenes



(Friedman and Farmer, 2018). Propionic and butyric acids are also other important organic acids in this process (Nah et al., 2018a; Nah et al., 2018b; Chebbi and Carlier, 1996). These low molecular weight monoacids have a vapor pressure of 6.8 ~ 8.1 µg m$^{-3}$ (log$_{10}$(C$^*$)) (Friedman and Farmer, 2018), and are generally considered to be too volatile to be distributed substantially to the particle phase. However, their presence in the aerosol particles is ubiquitous in

various areas over the world, although the levels of these monoacids are one or two order of magnitude lower than those in the gas phase (Nah et al., 2018a; Fisseha et al., 2004; Chebbi and Carlier, 1996). A study conducted in a forest-agriculture area in Atlanta showed that the acetic acid and formic acid are the second and third richest water-soluble organic acids in the particle phase and their molar fractions to the total individual acid concentrations in the particulate phase were 5.8 ± 5.0% and 3.6 ± 3.6%, respectively (Nah et al., 2018a). In addition, the presence of NH$_3$

as a strong base facilitates the shift of the equilibrium of these monocarboxylic acids and NH$_3$ to the particle phase (Barsanti et al., 2009). Under a base environment, a higher molar fraction of formic acid and acetic acid has also been observed in the particle phase (Nah et al., 2018a). These organic acids could exist in the particle phase in chemical forms of ammonium formate, ammonium acetate and ammonium propionate salts (Barsanti et al., 2009; Smith et al., 2008; Becker and Davidson, 1963), and thus contribute to the observed ammonium ions.

In an obvious contrast to the observation of C1-C5 monoacids above, the pinonic acids didn't show a linear correlation with the ammonium at the later stage of the nucleated SOA experiments (light pink region, Fig. 6d1), and there was no connection in the seeded SOA experiments. Pinonic acid is a typical first-generation marker product in the photooxidation reaction of a-pinene (Hallquiest et al., 2009). Na et al. (2007) reported that the addition of NH$_3$ to the aerosolized pinonic acid in an environmental reactor led to dramatic increase in both number and volume

concentrations of pinonic acid. They attributed the observation to the reaction between NH$_3$ and pinonic acids. However, the lack of positive correlation between the pinonic acid and ammoniums can be caused by several reasons in this study. Firstly, the vapor pressure of pinonic acid is about 4 ~ 5 orders of magnitude lower than those of C1-C5 monoacids (Jimenez et al., 2009), so pinonic acid can condense on the particle phase independent on NH$_3$. Meanwhile, the concentration of pinonic acid accounts for less than 1% of the total mono-acids based on the PTR-

MS results. Hence, the contribution of pinonic acid to the formed ammonium was estimated to be less than 1%, considering that the acidity strength of pinonic acid is similar to other C1-C5 monoacids (acid dissociation constant (pKa) of pinonic acid is 4.8 (Howell and Fisher, 1958), close to the pKa values of formic acid (3.75), acetic acid (4.75), propionic acid (4.86), butyric acid (4.83) and pentanoic acid (4.84) (Lide, 2007)). Therefore, after α-pinene was completely consumed and pinonic acid formation had ceased in the gas phase (pink region, Fig. 6), the rapid

condensation of pinonic acid on the particle phase or chamber wall causes a non-linear correlation of pinonic acid to ammonium. Secondly, the gas-phase pinonic acids could be further reacted away by OH radicals, which also contributes to the non-linear observation. Field experiment has shown the relatively low atmospheric PM concentrations of pinonic acid measured in summer because of the consumption of pinonic acid by OH radicals (Szmigielski et al., 2007).

Analogously, the scattering plots for the gas-phase organic dicarboxylic acids and particle-phase ammonium salts are shown in Fig. 7. We chose malonic acid and succinic acid as representative organic diacids, two of the most abundant dicarboxylic acids measured in the atmospheric aerosols (Chebbi and Carlier, 1996). The nice correlations of the gas-phase malonic and succinic acids to the particle-phase ammonium in this study suggest that diacids contribute to the formation of ammonium in both nucleated and seeded SOA experiments.

Our results qualitatively demonstrate that in the photooxidation of α-pinene, the presence of NH$_3$ drives the gas-phase mono- and dicarboxylic acids to the particle phase and promotes the SOA mass concentration in the CCN size.



Previous studied have shown that the presence of $NH_3$ can significantly enhance SOA formation from the a-pinene/ozone/photooxidation system because of the interaction of $NH_3$ with gas-phase organic acids (Na et al., 2007; Babar et al., 2017), which is consistent with our results. Carboxylic acid is one of the key species in determining SOA

physico-chemical property. Our results may prompt us to reconsider the pathway of gas phase organic acids involving in SOA formation in the atmosphere, whether they directly participate partitioning between the gas and particle phases, or they undergo secondary conversion via reaction with $NH_3$ before they condensate on the particle phase in the atmosphere. After SOA are formed, carbonyl group of chemical compounds in SOA particles can also uptake $NH_3$ heterogeneously to form nitrogen-containing compounds (Zhu et al., 2018; Liu et al., 2015; Updyke et al., 2012;

Dinar et al., 2007) and organic ammonium salts (Schlag et al., 2017). However, in this study the N:C ratios measured by AMS remained nearly constant at 0.002 for E0322 and E0326 and 0.004 for E0327 suggesting that the carbonyl-$NH_3$ heterogeneous reaction could be negligible. The interaction of $NH_3$ and SOA affects the cloud condensation nuclei (CCN) and hygroscopic growth of SOA particles and may have a potential impact on climate change (Dinar et al., 2007).

**3.3 Ammonium relevant to PMF-solved SOA component**

PMF analysis on the high-resolution organic mass spectra resolved the organic component to two factors in the nucleation experiments: MO-OOA (more oxidized oxygenated organic aerosol) and LO-OOA (less oxidized OOA). The oxidation level of MO-OOA is represented by an O:C ratio of 0.47 in this study, which is close to the value of 0.48 for MO-OOA determined in New York City in Summer (Sun et al., 2012). Its time series shows a good

correlation to the measured gas-phase monocarboxylic and di- acids such as butyric and succinic acids. The second factor LO-OOA is featured by an O:C ratio of 0.38. Its time series is related to the gas-phase oxidant products such as pinonaldehyde. The mass spectra profiles and time series of the two factors are shown in Fig. S3. In this study, the formation of ammonium salts is consistent with MO-OOA factor and also organic mono- and di- acids (Fig. 8). A higher oxidized organic factor is usually associated with the formation of organic mono- and di-acids. These results

further suggest that the ammonium has a close relation to the organic acids. Our observation is in agreement with the study by Schlag et al. (2017) where they showed by field data that $NH_4$ is associated with a more oxygenated organic aerosol factor.

**4 Conclusions**

The SOA experiments were carried out from photooxidation reaction of α-pinene in the presence of $NH_3$ in a

$29m^3$ indoor simulation chamber. Experiments were designed for SOA formation in the presence of ammonium sulfate seeds and at the absence of seed aerosols. The chemical composition and time-series of compounds in the gas- and particle-phase were characterized by an on-line high-resolution time-of-flight proton transfer reaction mass spectrometer (PTRMS) and a high-resolution time-of-flight aerosol mass spectrometer (AMS), respectively.

After the precursor α-pinene was consumed in the chamber, the mass concentration of organic aerosol in CCN-

size was decreased because of aerosol wall deposition or evaporation, the ammonium concentration was still rising, suggesting the continuous formation of ammonium. AMS results showed that organic acids were required to neutralize the observed ammonium salt. The $CO_2^+$ ion was selected to represent organic acids. The amount of $CO_2^+$ required for neutralizing ammonium accounted for the $27.0 \pm 3.1$ % of total $CO_2^+$ mass in the nucleated SOA experiments, and $18.7 \pm 6.0$ % in the seeded SOA experiments. The good correlation of organic monocarboxylic

acids (such as formic acid, acetic acid, propionic acid, butyric acid and pentanoic acid (or their isomers)) in the gas





phase to the ammonium salts further qualitatively confirms an affective role of organic acids for the ammonium formation. The same conclusion is also applied to the organic dicarboxylic acids such as malonic and succinic acids. In addition, the formed ammonium salts correlated well to the more-oxidized oxygenated organic aerosol (MO-OOA), which is consistent with the conclusion that organic acids contributed to the observed particulate ammonium.

Our work firmly shows the direct contribution of $NH_3$ to the CCN-size SOA formation through the organic acids-base reaction. The increase in SOA mass and the change of chemical composition due to $NH_3$-SOA interaction could change the hygroscopocity, CCN ability and optical property of aerosol particles, which may alter the aerosol impact on climate change and need to be studied in the future.

**Code/Data availability.**

The data included in this paper can be obtained by contacting the authors.

**Author contribution.**

LH, EK, AL and AV designed and conducted the experiments. LH and EK performed the data analysis with contributions by DW and AV. LH and AV wrote the paper with contributions from all co-authors.

**Competing interests.**

The authors declare that they have no conflict of interest.

**Acknowledgements**

The authors thank Mr. Ilkka Summanen for assisting the chamber experiments. This work was supported by The Academy of Finland Center of Excellence programme (grant no 307331), the European Research Council (ERC Starting Grant 335478) and EUROCHAMP-2020 Infrastructure Funding (grant no 730997).

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




Table 1 Initial experimental parameters and results in two sets of experiments

| | Exp. ID | a-Pinene (ppb) | $NO_x$ (ppb)[1] | R.H. (%) | Temp (°C) | AS seed surface[1] ($\mu m^2/cm^3$) | OH exposure (#/$cm^3$ s) | $\Delta NH_4$[2,3] mass ($\mu g\ m^{-3}$) | SOA [3] mass ($\mu g\ m^{-3}$) |
|---|---|---|---|---|---|---|---|---|---|
| Nucleation SOA Exp. | E0322 | 91.7 | bg (0.5) | 67.2 | 19.0 | bg | 1.2e11 | 1.2 | 391.8 |
| | E0326 | 100.9 | bg (0.3) | 50.5 | 21.4 | bg | 1.3e11 | 1.5 | 389.7 |
| | E0327 | 107.2 | 13.3 | 51.4 | 21.5 | bg | 1.3e11 | 1.6 | 476.1 |
| Seeded SOA Exp. | E0314 | 4.1 | 61.3 | 56.2 | 22.6 | 3.6e7 | 1.2e11 | 0.02 | 5.5 |
| | E0315 | 4.2 | bg (2.6) | 57.2 | 22.1 | 4.0e7 | 1.2e11 | 0.02 | 7.5 |
| | E0316 | 4.8 | bg (1.1) | 56.5 | 21.8 | 2.4e7 | 1.3e11 | 0.03 | 7.7 |

[1] bg=background concentration inside the chamber.

[2] In the seeded SOA experiment, because of the presence of ammonium sulfate seeds, the maximum mass concentration of newly formed ammonium salt was estimated from the difference between $NH_4^+{}_{,pre}$ and $NH_4^+{}_{,mea}$, refer to the text for details.

[3] The formed aerosol mass at maximum.










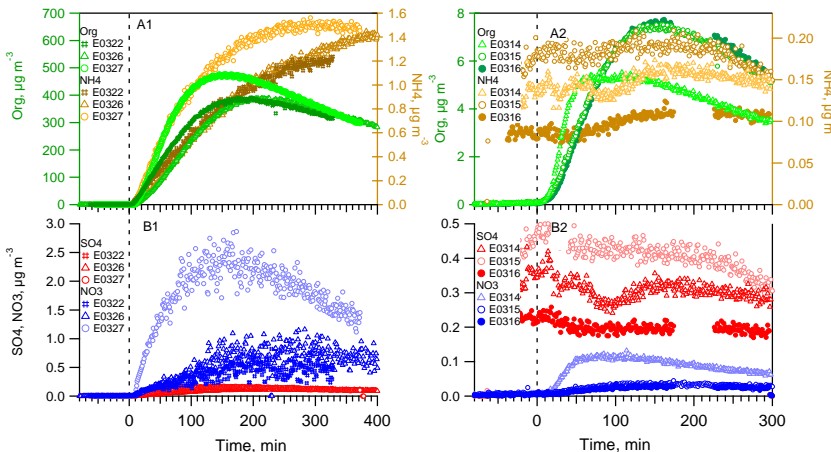


**Figure 1**. The mass concentrations of organics (left axis), ammonium (top panels, right axis), sulfate and nitrate (bottom panels) as a function of irradiation time in nucleation experiments (left panels) and in seeded SOA experiments (right panels). The irradiation time of 0 min marks the start of photooxidation reactions after UV lights were switched on.











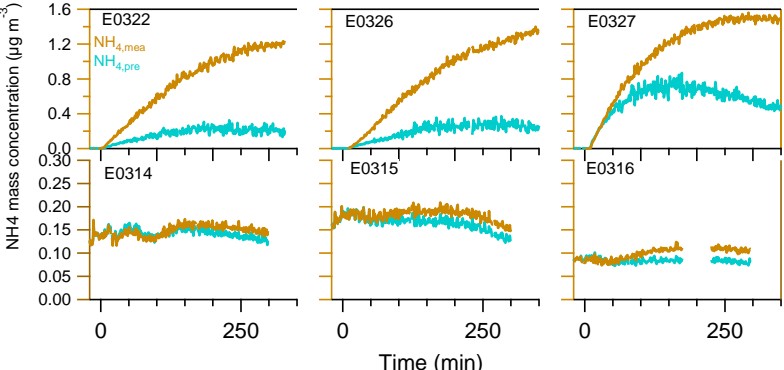

**Figure 2**. The time series of measured and predicted ammonium in the nucleated SOA (top panels) and in the seeded SOA experiments (bottom panels).





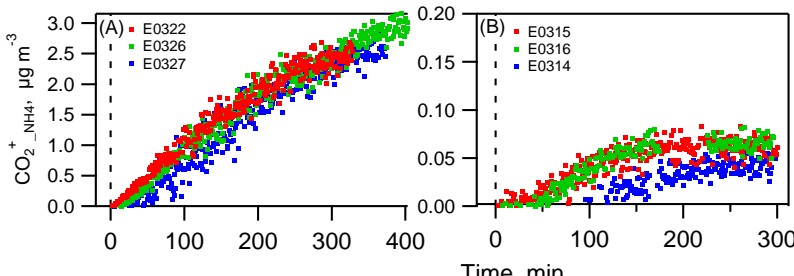

**Figure 3**. The estimated amount of organic carboxylic acid needed to fully neutralize SOA particles ($CO_2^+{}_{\_NH4}$). $CO_2^+$
group is chosen to represent organic acids.  (A) nucleation experiments (B) SOA seeded experiments. The time zero
marks the beginning of photooxidation reactions when UV lights were switched on.









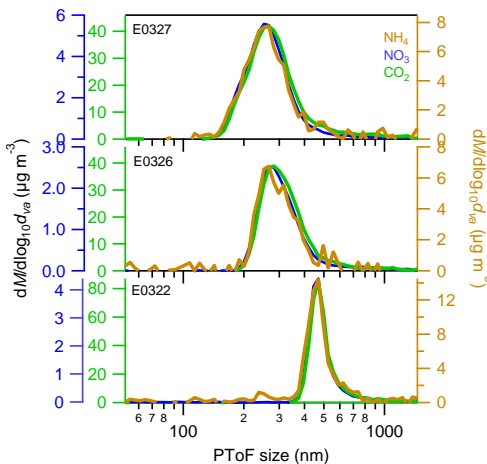

**Figure 4**. The averaged 30-minute size distribution of organic acids (represented by $CO_2^+$ ion measured by AMS), nitrate and ammonium in the last 30 minutes prior to the end of the nucleated SOA experiments.











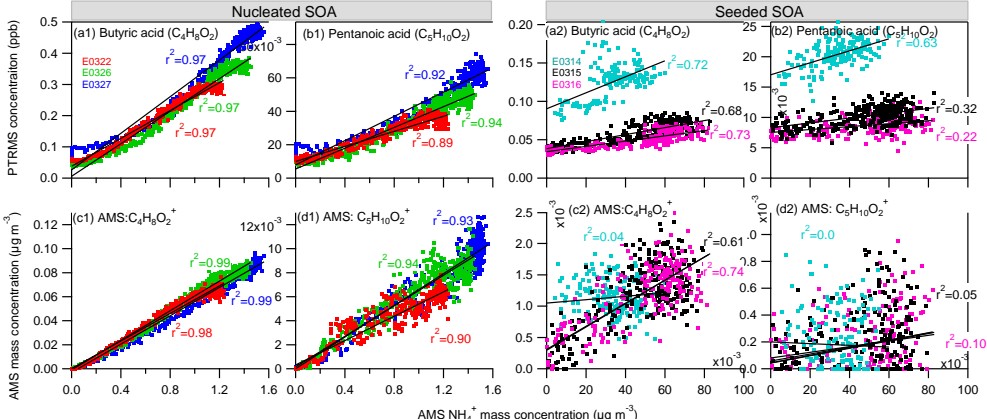

**Figure 5**. The comparison of measured compounds by PTRMS and by AMS in the nucleated SOA (left four panels) and seeded SOA experiments (right four panels). Top panels: Linear correlation between butyric ($C_4H_8O_2$) and pentanoic ($C_5H_{10}O_2$) monoacids measured by PTRMS measured and the ammonium by AMS; Bottom panels: Linear correlation of two fragmental ions $C_4H_8O_2^+$ and $C_5H_{10}O_2^+$ to the ammonium measured by AMS. Note that the two fragmental ions have identical ion molecular formula to those of butyric and pentanoic acids. In the seeded SOA experiments, the AMS $NH_4^+$ is the difference between $NH_4^+{}_{,pre}$ and $NH_4^+{}_{,mea}$, refer to the text for details.












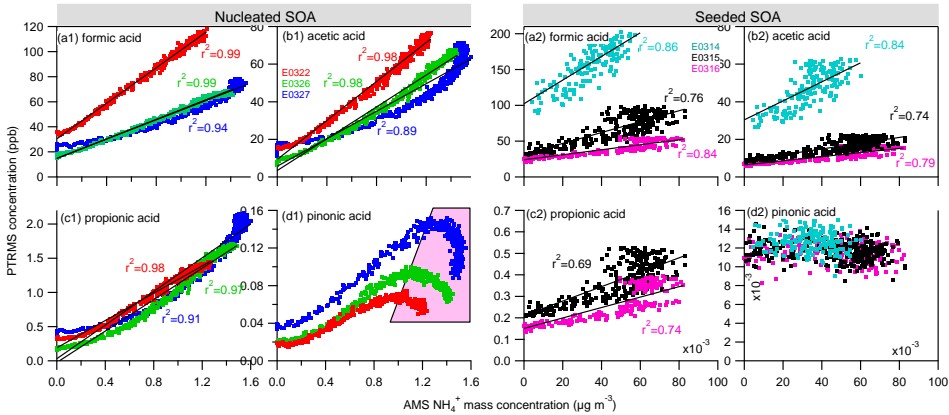

**Figure 6**. Correlations of the measured ammonium ion in the particle phase by AMS to the (a) formic acid, (b) acetic acid, (c) propionic acid, (d) pinonic acid in the gas phase measured by PTRMS in the nucleated SOA experiments (left four panels) and in the seeded SOA experiments (right four panels). The light pink area in panel d1 indicates a non-positive correlation regime between two species.












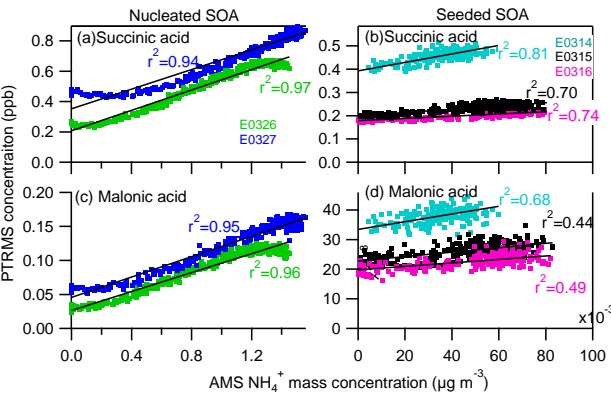

**Figure 7**. Correlations of the measured ammonium ion in the particle phase by AMS to two organic dicarboxylic
acids: (a) malonic acid, (b) succinic acid in the gas phase measured by PTRMS. Both of the diacids were not observed
in the experiment E0322.





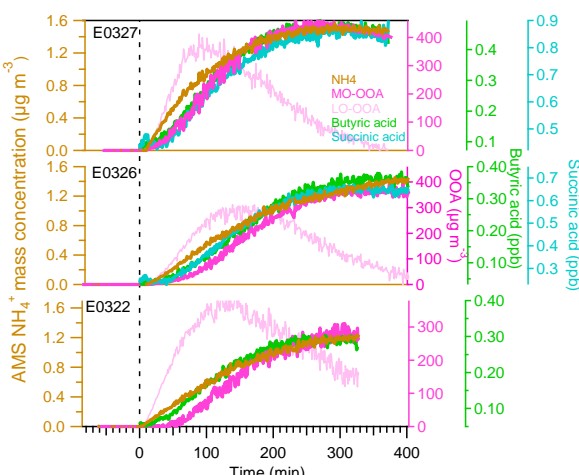

**Figure 8**. The relevance of particulate ammonium salt to the resolved MO-OOA component by PMF, and to the gas-phase butyric monocarboxylic acid and succinic diacid.