# Peer review of "Direct contribution of ammonia to alpha-pinene secondary organic aerosol formation"

_Atmospheric Chemistry and Physics, 2020_

## Author Comment (AC1) · 14 May 2020

We thank the editor for the nice comments.

Comment: Could you explain how NH3 was introduced into the chamber?

Reply: The NH3 is present as a background gas in our chamber, and we didn't specifically add it to the chamber. We will clarify this point in the revised manuscript.

Comment: Also I'd suggest showing the temporal evolution of NOx and SO2 in the supplement.

Reply: We have plotted the time series of NOx and SO2 shown in the attached figure. We will add them in the supplement.

[Figure]

[Figure]

[Figure]

**Figure S1**. The temporal evolution of NO$_x$ and SO$_2$ in the chamber during the nucleated and seeded SOA experiments.

**Fig. 1.**

---

## Referee Comment (RC1) · Anonymous Referee #1 · 19 Jun 2020

This manuscript reports new findings on the role ammonia in the formation of SOA from photooxidation of alpha-pinene. Neutralization of carboxylic acid by ammonia in the gas phase is reported as a process of CCN-sized SOA formation. PTR-MS and AMS were mainly used to measure relevant species in gas and particle phase, respectively, and to interpret the measured data for the neutralization. Although this manuscript is unique and concise, there are several main questions to be clarified. 1) In this manuscript key data used to support the process are correlations between organic acids and particulate ammonium. Mixing ratio of ammonia might be a critical factor in the photooxidation of smog chamber. It is not clear about the introduction to a rector bag and mixing ratio of ammonia in the bag. The authors report N/C ratio remained consistent, implying negligible formation of organonitrogen via particle phase

reaction. Reaction pathways might be largely dependent on the mixing ratio of ammonia. 2) For seeded experiments alpha-pinene concentration was too low compared to nucleation experiments. It is uncertain why such low concentration was used for seeded cases. It makes difficult to clearly show the temporal variation of organic acids. In seeded experiments correlation of organic acids with ammonium are not apparent as much as in nucleation experiments, probably due to lower concentration of alpha-pinene. A result stated in P5, line 181-183 (48.6 times lower concentration of $CO_2+$ for seeded experiment) might be just due to lower initial ïĄą-pinene concentration for seeded condition (23 times lower). 3) Figure 3 apparently show the difference in a lag time of $CO_2+$_NH4. It was partly explained that nitric acid delayed its accumulation in high NOx condition. More explanation might be necessary for the delay in low NOx conditions. Very limited amount of $CO_2+$_NH4 was formed roughly at 1/100 of SOA in mass both in nucleation and seeded experiments. Is it caused by limit in the available low MW carboxylic acids, ammonia, or other factors? It is probably associated with the contribution of neutralization and its atmospheric implication due to the omnipresence of NOx and nitric acids. 4) Although a previous paper (Friedman and Farmer, 2018) also reported the formation of low molecular weight organic acids as observed in this manuscript, little information is available about the formation mechanism of those species. Even a brief introduction of the formation mechanism would be very useful in the understanding of neutralization process. Overall, I would recommend the publication of the manuscript if the authors can address my questions and comments. Minor comments are show below.

P3, line 89-91: H2O2 was introduced as a source of OH radical. Although OH exposure is presented in Table 1, H2O2 concentration itself is informative for readers.

P5, line 173, 179: It is curious why chloride ion was added in the estimation of NH4+, since there was no source of chloride in the smog chamber experiments.

P6, Line 228: "Fig. 4" might a typo of "Fig. 6".

[Figure]

P7, line 270-274: Diacids such as malonic and succinic acids have vapor pressure similar to pinonic acid. Differently from pinonic acid, those diacids showed good correlation with ammonium. This might mean continuous formation of those diacids. It is not clear how these nonvolatile diacids could form in the gas phase until the later part of photooxidation.

P9, Line 316: "affective" might be a typo of "effective".

P15, Table 1: a-Pinene might be a typo of "alpha-Pinene".

P18, Figure 3: In the caption, "SOA" needs to be deleted in "... (B) SOA seeded experiments ..."

P20, Figure 5: In the caption, "the AMS NH4+ is the difference between NH4+,pre and NH4+,mea, refer to the text for details." needs to be clarified. In the text NH4+,pre and NH4+,mea are defined, whereas "AMS NH4+" is not defined. It should be defined clearly in the caption, e.g., NH4+,mea - NH4+,pre. It should be clarified in the caption of figures 6 and 7. It might be better to move the position of "x10-3" to improve the readability in the figure. It is same in figures 6 and 7.

P21, Figure 6: Readers might expect to see plots of AMS data also as Figure 5.

P22, Figure 7: Readers might expect to see plots of AMS data also as Figure 5.

It is worthy to clearly note in the text that all particulate data were not corrected to wall loss of particulate.

In the figure, axis titles need to be checked to properly note "NH4+", "SO42-", and "NO3-".

---

## Referee Comment (RC2) · Anonymous Referee #2 · 26 Jun 2020

This manuscript reports chamber results of alpha-pinene SOA formation in the presence of ammonia, and suggested that organic acids have a central role in the formation of particle phase ammonium through neutralization. Major arguments supporting this conclusion come from the stoichiometric neutralization analysis, and correlations between gas-phase organic acids and particle-phase NH4+. The experiment looks carefully conducted and calibrated, with interesting results. However, the interpretation of the results is ambiguous and confusing. Although I trust in the data, I'm not convinced by the logic that draws to the conclusions. The following concerns should be addressed before this work can be considered for publication for ACP.

Major concerns

1. How did the authors estimate the NH3 concentration in the chamber "by assuming

that the particulate ammonium salt (NH4+) was converted from the gas-phase NH3 (Fig. 1)"? Did the authors assume a 1:1 conversion ratio, namely all gas-phase NH3 are converted to NH4+? If so, this assumption is too arbitrary as NH3 is highly volatile. If the authors are introducing the NH3 from the "background gas", I'd suggest at least an estimate of the typical NH3 level (e.g., through a supplementary measurement of typical background NH3 concentration in your lab, or provide the measured NH3 level in the vicinity environments), and do a calculation to check whether all the NH3 would have been depleted by the organic acids at your measured concentration level.

2. What on earth is the definition of CCN size in this paper?

3. The results seem to suggest that the measured NH4 concentration is more than enough to neutralize the inorganic acids (SO4, NO3 and Cl) (Eq. 1), while not enough to neutralize all organic acids (CO2+), as "the amount of CO2+ required for neutralizing ammonium accounted for the 27.0 ± 3.1 % of total CO2+ mass in the nucleated SOA experiments, and 18.7 ± 6.0 % in the seeded SOA experiments". If so, then ammonia would be the limiting species, the concentration of which should be determined by available total ammonia concentrations. In this case, it is confusing why the NH4+ would correlated to gas-phase organic acids. If the authors don't really think the CO2+ all comes from organic acids, it should be clarified in the manuscript.

4. Following above question, the good correlations among gas-phase organic acids and NH4+ should be expected if the gas-phase organic acids are the limiting species, with total ammonium more than enough to influence the partitioning equilibriums. The results seem self-conflicting.

5. The authors argue that "The reaction of HNO3 and NH3 takes precedence over the reaction between organic acids and NH3" (Line 186). Fig. 1 also shows formation of sulfate and nitrate from photooxidation of SO2 and NOx. If so, the influences from inorganics should be first excluded in analyzing the correlations of NH4+ and gas-phase organic acids (Fig. 5). Instead of AMS NH4+, the difference between measured

NH4+ and that predicted by stoichiometric neutralization analysis of inorganic acid (Eq. 1) should be used (i.e., Free NH4+ = NH4+,meas - NH4+,pre). This is similar concept with the correction of (NH4)2SO4 for the seeded experiment in Fig. 5. How would the correlation look like after this kind of correlation?

Minor concerns

1. There're some typos in the manuscript. For example, Line 59, "updake" should be "uptake". Line 131, "aftert" should be "after". Line 135, there's duplicate periods. The manuscript should be read through more carefully.

2. I'd suggest name the predicted NH4+ differently for that based on Eq. 1 and Eq. 2, i.e. that predicted with / without consideration of CO2+.

---

## Author Comment (AC3) · 21 Aug 2020

**Responses to Referee 2:**

The authors thank the reviewer for her/his excellent comments. We have modified our manuscript according to the reviewer's comments. All the changes that we have made were saved as a format of "Track Changes" in the manuscript.

*This manuscript reports chamber results of alpha-pinene SOA formation in the presence of ammonia, and suggested that organic acids have a central role in the formation of particle phase ammonium through neutralization. Major arguments supporting this conclusion come from the stoichiometric neutralization analysis, and correlations between gas-phase organic acids and particle-phase NH4+. The experiment looks carefully conducted and calibrated, with interesting results. However, the interpretation of the results is ambiguous and confusing. Although I trust in the data, I'm not convinced by the logic that draws to the conclusions. The following concerns should be addressed before this work can be considered for publication for ACP.*
Reply:
We thank the referee for the positive comments.

Major concerns
*1. How did the authors estimate the NH3 concentration in the chamber "by assuming that the particulate ammonium salt (NH4+) was converted from the gas-phase NH3 (Fig. 1)"? Did the authors assume a 1:1 conversion ratio, namely all gas-phase NH3 are converted to NH4+? If so, this assumption is too arbitrary as NH3 is highly volatile. If the authors are introducing the NH3 from the "background gas", I'd suggest at least an estimate of the typical NH3 level (e.g., through a supplementary measurement of typical background NH3 concentration in your lab, or provide the measured NH3 level in the vicinity environments), and do a calculation to check whether all the NH3 would have been depleted by the organic acids at your measured concentration level.*

Reply:
   We would like to highlight that our estimation gives the lowest boundary for the possible $NH_3$ concentration. Hence, in our estimation we indeed assumed a conversion ratio of 1:1.
   The ammonia was unintentionally introduced to the chamber quite likely with the pressurized air and water used for humidification. Unfortunately, we lacked the ammonia measurement device in our lab. Based on a statistics study by Salonen et al (2009) on the ammonia in 14 Finnish office buildings where there were no specific emission source, the indoor concentrations varied between 1-49 ug m$^{-3}$ with a geometric mean concentration of 14 ug m$^{-3}$ (corresponds 19 ppbV). If we assume that ammonia concentrations in our experiments are on the same order with the highest value reported in Salonen et al. study, we can assume that in the beginning of photooxidation reaction after UV lights were switched on, the ammonia concentration is higher than organic acid concentration, and organic acids should be the limiting species. As photooxidation reactions went on and organic acids formation accumulated in the chamber, their concentration was higher than ammonia, and the ammonia was a limiting species and would be neutralized by the organic acids at the end. However, it is still an open question about the distribution of organic ammonium: how much they are present in the gas phase and how much in the particle phase?
   Accordingly, we have added in line these in line92, page 3 that "In two sets of experiments, ammonia was introduced to the chamber as an impurity. The likely sources of ammonia were the pressurized air, possible leakage of the lab air and water used for humidification".
   And also in line 126, we added that "The maximum $NH_4^+$ concentration was in the range of 1.17~1.51 µg m$^{-3}$, which corresponds to a minimum $NH_3$ concentration level of 1.6 ~ 2.1 ppbV in our chamber. The $NH_3$ mixing ratio over continental range is typically between 0.1 and 10 ppb (Seinfeld and Pandis, 2016). A statistics study on the ammonia concentration in 14 Finnish office buildings shows a range of 1-49 ug m$^{-3}$ and a geometric mean concentration of 14 ug m$^{-3}$ (corresponds 19 ppbV) (Salonen et al., 2009). Hence, our method should provide a lowest boundary of ammonia mixing ratio in the chamber."

*2. What on earth is the definition of CCN size in this paper?*

Reply:

The CCN size naturally depends on particle composition, but in general, it is often assumed that atmospheric aerosol particles larger than 100 nm are able to active to cloud droplets. AMS measurement range is 35-1000 nm, and the aerosol mass is centered on the larger sizes, hence we can state that our measurements are in CCN size range. We have now clarified this in line 145, page 4 that "As the measured mass is centered on the larger sizes of AMS measurement range (35-1000nm, Jayne et al., 2000; Zhange et al., 2004), we can state that the measurements represents CCN relevant particle sizes (in general it is often assumed that atmospheric aerosol particles larger than 100 nm are able to active to cloud droplets)."

*3. The results seem to suggest that the measured NH4 concentration is more than enough to neutralize the inorganic acids (SO4, NO3 and Cl) (Eq. 1), while not enough to neutralize all organic acids ($CO_2^+$), as "the amount of $CO_2^+$ required for neutralizing ammonium accounted for the 27.0 _ 3.1 % of total $CO_2^+$ mass in the nucleated SOA experiments, and 18.7 _ 6.0 % in the seeded SOA experiments". If so, then ammonia would be the limiting species, the concentration of which should be determined by available total ammonia concentrations. In this case, it is confusing why the $NH_4^+$ would correlated to gas-phase organic acids. If the authors don't really think the CO2+ all comes from organic acids, it should be clarified in the manuscript.*

Reply:

We thank the referee for the excellent comment and the next one. The referee is correct that $NH_4^+$ is sufficient to neutralize the inorganic acids, but not adequate to neutralize organic acids, and it is a limiting species, as we have speculated in the reply to the first general comment.

We would first make it clear that in our study the $NH_4^+$ is observed to correlate to the C1-C5 monoacids and two diacids, but not all organic acids, e.g. no correlation to pinonic acid. Fig. C1 shows time series of the particle-phase total $CO_2^+$_total (green), the ammonium resulting from organic acids ($NH_4^+$_orgacid) , and gas-phase formic acid and pinonic acid in the nucleated SOA experiments. The formic acids was selected to represent the low molecule-weight (MW) organic monoacids, and pinonic acid (molecule weight is 184) was chosen to represent high MW organic acid. We can see that:

(1) The time series of the total particulate organic acids ($CO_2^+$_total in Fig. C1) is not following the low MW organic acids (formic acid), nor the high MW organic acids (pinonic acid), suggesting that particulate organic acids were not derived from one single type of gas organic acids.

(2) The formation of $NH_4^+$_orgacid is mainly related to the low MW organic monoacids. We can see that the trend of formic acid formation is consistent with $NH_4^+$_orgacid. In the experiments, we observed a high production of these low MW organic monoacids: their concentrations were about one or two orders of magnitude greater than the concentration of high MW acids. Hence, compared with the high MW acids, the low MW monoacids dominated the reaction with ammonia in the gas phase if we assume similar reaction rate constants $k_{orgacid}$ for both low- and high- MW acids with ammonia ($R=k_{orgacid}·[OrgAcid]·[NH_3]$), and the majority of $NH_4^+$_orgacid are associated with the low MW monoacids. Therefore, $NH_4^+$_orgacid is well correlated to the low-MW organic monoacids. We need to point out that although the low MW organic monoacids dominate the ammonia reaction, the formed ammonium salts are still volatile, so we speculate that a large amount of them are present in the gas phase.

(3) Inconsistency between the formation of high MW organic acids and the $NH_4^+$_orgacid. Here we use pinonic acid as an example because it is a high MW organic acid that has been measured in our work. The vapor pressure of pinonic acid is about 4-5 orders of magnitude lower than those of C1-C5 monoacids (Jimenez et al., 2009), and its reactivity to ammonia is more than 500 times lower than that of C1-C5 monoacids. This allows pinonic acid to condense directly on the particle phase before it reacts in large amount with ammonia. These condensed high-MW organic acids are efficient to fragment to form $CO_2^+$ in AMS, and produced more than 73.0 % and 81.3% of the observed $CO_2^+$ in the nucleated and seeded SOA experiments, respectively. The remaining $CO_2^+$ signals (less than 27.0 % and 18.7 %) came from the interaction between low MW organic acids and ammonia. Our analysis is also consistent with other studies showing that the $CO_2^+$ detected in AMS is caused by the thermal decomposition of the mono-, di-, and poly carboxylic acid groups, and is related to high MW oxygenated organic species (e.g. Zhang

et al., 2005; Alfarra et al., 2004). We still want to point out that the $CO_2^+{}_{\_total}$ accounts for 2.7% to 7.0% of SOA mass at maximum in each experiments, and it is very possible that the high-MW organic acids contributed to the majority of these $CO_2^+{}_{\_total}$.

So, our above analysis interprets the observation that $NH_4^+{}_{\_orgacid}$ correlates well to the low MW organic monoacids, even $CO_2^+{}_{\_NH4}$ accounts for only 27.0 % and 18.7 % of the total $CO_2^+$.

[Figure]

Figure C1. The evolution of time series of the particle-phase total $CO_2^+{}_{\_total}$ (green), the formed ammonium attributed to organic acid neutralization ($NH_4^+{}_{\_orgacid}$), and gas-phase formic acid and pinonic acid in the nucleated SOA experiments. Formic acid is chosen to represent C1-C5 organic monoacids and pinonic acid as a high molecule-weight organic acid.

*4. Following above question, the good correlations among gas-phase organic acids and NH4+ should be expected if the gas-phase organic acids are the limiting species, with total ammonium more than enough to influence the partitioning equilibriums. The results seem self-conflicting.*
Reply:
As we have replied to the prior comment, ammonia is the limiting species in our experiments. The majority of $NH_4^+{}_{\_orgacid}$ are associated with the low MW monoacids, and the high-MW organic acids contributed to the majority of the observed $CO_2^+$. Our analysis is consistent with the observation that $NH_4^+{}_{\_orgacid}$ correlates well to the low MW organic monoacids.

*5. The authors argue that "The reaction of HNO3 and NH3 takes precedence over the reaction between organic acids and NH3" (Line 186). Fig. 1 also shows formation of sulfate and nitrate from photooxidation of SO2 and NOx. If so, the influences from inorganics should be first excluded in analyzing the correlations of NH4+ and gas phase organic acids (Fig. 5). Instead of AMS NH4+, the difference between measuredNH4+ and that predicted by stoichiometric neutralization analysis of inorganic acid (Eq.1) should be used (i.e., Free NH4+ = NH4+,meas - NH4+,pre). This is similar concept with the correction of (NH4)2SO4 for the seeded experiment in Fig. 5. How would the correlation look like after this kind of correlation?*
Reply:
Following the referee's comment, in the nucleated SOA experiments, the sulfate, nitrate and chloride related ammonium was excluded from the $NH_4^+{}_{\_mea}$, so only organic acid-related ammonium was used for plotting in Figs. 5-7. The identical analysis protocol was applied in both nucleated and seeded SOA

experiments. After we did the analysis suggested by the referee, the correlation coefficients don't change much (table C1). Meanwhile, we also noticed an error in the seeded SOA experiments, that is, we misused ammonium-related organic acids for X-axis in Figs. 5-7; Instead, the correct one should be organic acids-related ammonium. The mistake doesn't affect any of our conclusion, but after correction, the correlation relationship is becoming better in the figures.

Table C1. The correlation coefficients in the new and old figures.

| Correlation species | Correlation coefficient ($r^2$) | | | |
|---|---|---|---|---|
| | Nucleated SOA(old) | Nucleated SOA(new) | Seeded SOA(old) | Seeded SOA(new) |
| Butyric acid vs $NH_4^+{}_{\_orgacid}$ (Fig.5) | 0.97-0.97 | 0.88-0.95 | 0.68-0.73 | 0.76-0.83 |
| $C_4H_8O_2^+$ vs $NH_4^+{}_{\_orgacid}$ (Fig.5) | 0.98-0.99 | 0.90-0.97 | 0.04-0.74 | 0.12-0.80 |
| Pentanoic acid vs $NH_4^+{}_{\_orgacid}$ (Fig.5) | 0.89-0.94 | 0.84-0.92 | 0.22-0.63 | 0.27-0.62 |
| $C_5H_{10}O_2^+$ vs $NH_4^+{}_{\_orgacid}$ (Fig.5) | 0.90-0.94 | 0.84-0.91 | 0.05-0.10 | 0.0-0.14 |
| Formic acid vs $NH_4^+{}_{\_orgacid}$ (Fig.6) | 0.94-0.99 | 0.95-0.97 | 0.76-0.86 | 0.81-0.87 |
| Acetic acid vs vs $NH_4^+{}_{\_orgacid}$ (Fig.6) | 0.89-0.98 | 0.97-0.98 | 0.74-0.84 | 0.80-0.84 |
| Propionic acis vs $NH_4^+{}_{\_orgacid}$ (Fig.6) | 0.91-0.98 | 0.96-0.97 | 0.69-0.74 | 0.72-0.79 |
| Succinic acid vs $NH_4^+{}_{\_orgacid}$ (Fig.7) | 0.94-0.97 | 0.91-0.94 | 0.70-0.81 | 0.51-0.81 |
| Malonic acid vs $NH_4^+{}_{\_orgacid}$ (Fig.7) | 0.95-0.96 | 0.79-0.90 | 0.44-0.68 | 0.42-0.67 |

Minor concerns

*1.There're some typos in the manuscript. For example, Line 59, "updake" should be "uptake". Line 131, "aftert" should be "after". Line 135, there's duplicate periods. The  manuscript should be read through more carefully.*

Reply:

The typos were corrected and we have read the manuscript carefully.

*2. I'd suggest name the predicted NH4+ differently for that based on Eq. 1 and Eq. 2, i.e. that predicted with / without consideration of CO2+.*

Reply:

We have changed $NH_4^+{}_{,pre}$ to $NH_4^+{}_{,pre\_CO2}$ in Eq. 2.

Reference

Alfarra, M.R., Coe, H., Allan, J.D., Bower, K.N., Boudries, H., Canagaratna, M.R., Jimenez, J.L., Jayne, J.T., Garforth, A.A., Li, S., Worsnop, D.R., Characterization of urban and rural organic particulate in the Lower Fraser Valley using two Aerodyne Aerosol Mass Spectrometers, Atmos. Environ., 38, 5745-5758, 2004.

Jayne, J. T.,  Leard, D. C., Zhang, X. F.,  Davidovits, P.,  Smith, K. A., Kolb, C. E., and Worsnop, D. R.:  Development of an Aerosol Mass Spectrometer for size and composition analysis of submicron particles, Aerosol Sci. Technol., 33, 49–70, 2000.

Jimenez, J.L., M.R. Canagaratna, N.M. Donahue, et al., Evolution of Organic Aerosols in the Atmosphere. Science, 326, 1525-1529, DOI: 10.1126/science.1180353, 2009.

Salonen, H., Pasanen, A., Lappalainen, S.K., Riuttala, H.M., Tuomi, T.M., Pasanen, P.O., Back, B.C. and Reijula, K.E., Airborne concentrations of volatile organic compounds, formaldehyde and ammonia in Finnish office buildings with suspected indoor air problems, Journal of Occupational and Environmental Hygiene, 6, 200-209, 2009.

Seinfeld, J.H. and Pandis, S.N., Atmospheric chemistryand physics: From air pollution to climate change, 3rd edition, John Wiley & Sons, Inc.,2016.

Zhang, Q., Alfarra, M.R., Worsnop, D.R., Allan, J.D., Coe, H., Canagaratna, M.R., and Jimenez, J.L.,Deconvolution and quantification of hydrocarbon-like and oxygenated organic aerosols based on aerosol mass spectrometry, Environ. Sci. Technol., 39, 4938-4952, 2005.

---

## Author Comment (AC2)

**Responses to Referee 1:**

The authors thank the reviewer for her/his excellent comments. We have modified our manuscript according to the reviewer's comments. All the changes that we have made were saved as a format of "Track Changes" in the manuscript.

*This manuscript reports new findings on the role ammonia in the formation of SOA from photooxidation of alpha-pinene. Neutralization of carboxylic acid by ammonia in the gas phase is reported as a process of CCN-sized SOA formation. PTR-MS and AMS were mainly used to measure relevant species in gas and particle phase, respectively, and to interpret the measured data for the neutralization. Although this manuscript is unique and concise, there are several main questions to be clarified.*
*1) In this manuscript key data used to support the process are correlations between organic acids and particulate ammonium. Mixing ratio of ammonia might be a critical factor in the photooxidation of smog chamber. It is not clear about the introduction to a rector bag and mixing ratio of ammonia in the bag. The authors report N/C ratio remained consistent, implying negligible formation of organonitrogen via particle phase reaction. Reaction pathways might be largely dependent on the mixing ratio of ammonia.*

Reply:
Both referrers are concerned about the introduction of ammonia and its concentration in this study. We would like to point out that ammonia was not intentionally introduced to the chamber, but it was present as an impurity. Likely sources of ammonia were pressurized air, possible leakage of lab air, and water used for humidification of the chamber. We have added these in line92, page 3 that "In two sets of experiments, ammonia was introduced to the chamber as an impurity. The likely sources of ammonia were the pressurized air, possible leakage of the lab air and water used for humidification".

As we have stated in the manuscript in line 124-128 that we unfortunately lacked the measurement of $NH_3$ concentration in the chamber but estimated the minimum concentration of $NH_3$ present from our AMS measurement results, by assuming that the particulate ammonium salt ($NH_4^+$) was converted from the gas-phase $NH_3$ (Fig. 1). The maximum $NH_4^+$ concentration was in the range of 1.17~1.51 μg $m^{-3}$, which corresponds to a minimum $NH_3$ concentration level of 1.6 ~ 2.1 ppbV in our chamber. The $NH_3$ mixing ratio over continental range is typically between 0.1 and 10ppb (Seinfeld and Pandis, 2016). A statistics study on the ammonia concentration in 14 Finnish office buildings shows a range of 1-49 ug $m^{-3}$ and a geometric mean concentration of 14 ug $m^{-3}$ (corresponds 19 ppbV) (Salonen et al., 2009). Hence, our method should provide a lowest boundary of ammonia mixing ratio in the chamber. We have added the discussion in page 4 in the manuscript.

According to Salonen et al., the detected highest ammonia concentration at Finnish office building is 49 ug $m^{-3}$ (corresponding to 64ppb). This gives an approximatively higher limit for the order of expected ammonia concentration that we can use to perform the following analysis. Based on PTRMS measurements, we estimated the formed organic acid concentration to be about 50-250ppb at the end of our experiments. After comparison of the concentrations to the two species, the ammonia was a limiting species and would be neutralized by the organic acids at the end.

The referee's comments also guide us to recheck the N/C ration in both nucleated and seeded SOA experiments (Fig C1). Regardless of the relatively high noise level of our data, we can see the nearly constant or small time dependent increase in N/C ratio in both sets of experiments. Therefore, line 285 page 8, we have reworded the statement "After SOA are formed, carbonyl group of chemical compounds in SOA particles can also uptake $NH_3$ heterogeneously to form nitrogen-containing compounds (Zhu et al., 2018; Liu et al., 2015; Updyke et al., 2012; Dinar et al., 2007) and organic ammonium salts (Schlag et al., 2017). However, in this study the N:C ratios measured by AMS remained nearly constant at 0.002 for E0322 and E0326 and 0.004 for E0327 suggesting that the carbonyl-$NH_3$ heterogeneous reaction could be negligible." to "After SOA are formed, carbonyl group of chemical compounds in SOA particles can also uptake $NH_3$ heterogeneously to form nitrogen-containing compounds (Zhu et al., 2018; Liu et al., 2015; Updyke et al., 2012; Dinar et al., 2007) and organic ammonium salts (Schlag et al., 2017). Similarly, in this study the N:C ratios measured by AMS

were nearly constant or slightly increasing in two sets of experiments (Fig. C1), suggesting the formation of nitrogen-containing compounds via carbonyl-NH₃ heterogeneous reaction.".

Definitely, we strongly agree with the referee that the accurate measurement of ammonia in the chamber is critical to understand the detailed mechanisms of neutralization of gas-phase organic acids, formation of particulate $CO_2^+$ and formation of NOC in the specific experiments.

[Figure]

Figure C1. The N:C ratios determined by AMS in (A) the nucleated SOA experiments, and (2) in the seeded SOA experiments.

*2) For seeded experiments alpha-pinene concentration was too low compared to nucleation experiments. It is uncertain why such low concentration was used for seeded cases. It makes difficult to clearly show the temporal variation of organic acids. In seeded experiments correlation of organic acids with ammonium are not apparent as much as in nucleation experiments, probably due to lower concentration of alpha-pinene. A result stated in P5, line 181-183 (48.6 times lower concentration of CO2+ for seeded experiment) might be just due to lower initial α-pinene concentration for seeded condition (23 times lower).*

Reply:

We agree with the referee that in the seeded experiments αlpha-pinene concentration was about 20 times lower than in the nucleation experiment. The relatively low αlpha-pinene concentration was adopted to mimic an atmospherically relevant monoterpene mixing ratio, for example, in Hyytiälä forest area in Finland (e.g. Kourtchev *et al.*, 2006). We aimed to study if the ammonia plays a similar role as in the high a-pinene concentration case. Meanwhile, in the presence of seed aerosol, the relative low a-pinene can produce detectable amount of secondary organic aerosol (SOA) for the aerosol mass spectrometer (AMS). As the referee pointed out, the correlation of organic acids with ammonium is less obvious in the seeded, low αlpha-pinene concentration experiments compared to the nucleation experiments. Part of reason is that in the seeded experiments, the produced ammonium resulting from organic acid neutralization ($NH_4^+{}_{orgacid}$) is around 40 times less than in the nucleated cases. Such small amount of $NH_4^+{}_{orgacid}$ (at about 0.03 ug m⁻³ level) made the determination to its concentration be noisy by AMS. Hence, its correlation to organic acids is worse in the seeded experiments than in the nucleated SOA experiments. Nevertheless, the results in two sets of experiments are consistent making our conclusion even stronger.

*3)*
*3.1) Figure 3 apparently show the difference in a lag time of CO2+_NH4. It was partly explained that nitric acid delayed its accumulation in high NOx condition. More explanation might be necessary for the delay in low NOx conditions.*

Reply:

A lag in time for $CO_2^+{}_{NH4}$ is also observed in the low $NO_x$ conditions, especially in the seeded experiments. The reasons for the observed delay is again affected by the nitric acid arising from the background $NO_x$ photooxidation, where we have observed particulate nitrate formation (Fig C2). The

other reason is that the formation of particulate organic acids was much slower in the seeded experiments compared to the nucleated cases (Fig C3), implying that their reaction with ammonia was also delayed.

Accordingly, we have added the argument in line 189 " In the low-$NO_x$ test, a time lag for $CO_2^+$_NH4 is also observed, especially in the seeded SOA experiments. The delay is caused by the effect of nitric acid arising from the background $NO_x$ photooxidation. In addition, the slower formation of particulate organic acids makes their reaction with ammonia delayed in the seeded experiments compared to the nucleated cases."

[Figure]

Figure C2. The formed nitrate aerosol measured by AMS in the seeded SOA experiment.

[Figure]

Figure C3. The measured $CO_2^+$ ion concentration measured by AMS (A) in the seeded and (B) in the nucleated SOA experiments. $CO_2^+$ ion was chosen to represent organic acids. The formation rate of $CO_2^+$ in seeded experiments was much slower than in the nucleated cases.

*3.2) Very limited amount of CO2+_NH4 was formed roughly at 1/100 of SOA in mass both in nucleation and seeded experiments. Is it caused by limit in the available low MW carboxylic acids, ammonia, or other factors? It is probably associated with the contribution of neutralization and its atmospheric implication due to the omnipresence of NOx and nitric acids.*

Reply:

In the two sets of experiments, the total $CO_2^+$ ions account for 2.7% to 7.0% of SOA in mass (Fig. C4). Out of the $CO_2^+$ ions, only $27.0 \pm 3.1$ % and $18.7 \pm 6.0$ % have participated in the reaction with ammonia in the nucleated and the seeded SOA experiments, respectively. This explains why $CO_2^+$_NH4 is roughly 1/100 of SOA in mass. In our experiments, the ammonia could be a limiting species. The produced low-MW carboxylic acid concentration is about one or two orders of magnitude greater than the high-MW acids. Hence, the reaction between low-MW organic acids and ammonia dominated over the reactions between high-MW acids and ammonia. The vapor pressure of the formed low-MW organic ammonium

salts is in the same order as their parental acids. Therefore, we speculate that a large amount of these low-MW organic ammonium salts were present in the gas phase and a minor amount of ammonium salts have participate on the particulate phase, contributing to $CO_2^+{}_{\_NH4}$. The amount of $CO_2^+{}_{\_NH4}$ is assumed to be related to: (1) the photooxidation and ozonolysis capacity in our chamber, which determines the amount of particulate organic acids (total $CO_2^+$), (2) the relative amount of low- and high- MW organic acids and their physicochemical properties, (3) the relative amount of inorganic and organic acids, and (4) the relative amount of ammonia and total acids. We have added the discussion in line 190 in the manuscript.

[Figure]

Figure C4. The mass fraction of total $CO_2^+$ ions to the SOA mass.

*4) Although a previous paper (Friedman and Farmer, 2018) also reported the formation of low molecular weight organic acids as observed in this manuscript, little information is available about the formation mechanism of those species. Even a brief introduction of the formation mechanism would be very useful in the understanding of neutralization process.*

Reply:

The formation of low molecular weight organic acids remains unclear in the photochemical reaction of $\alpha$-pinene (e.g.Friedman and Farmer, 2018). It can be speculated that these organic acids were produced as products of stabilized Criegee intermediate associated with ozonolysis of $\alpha$-pinene and double carbon bond-containing products (Jacob and Wofsy, 1988; Orzechowska and Paulson, 2005). The statement is verified by the fact that we have observed the highest $O_3$ formation at the highest $NO_x/VOC_x$ initial input in E0314 (Figure C5). In E0314, we also have observed higher concentrations of C1-C5 monoacids corresponding to higher ozonolysis reactivity (Figure C6). The specific formation mechanisms of the organic acids need to be investigated in the future studies. We have included the discussion and the figures C5 and C6 in the manuscript.

[Figure]

Figure C5. The $O_3$ and $NO_x$ concentration in the chamber. The high $NO_x$ injection led to high $O_3$ concentration in the chamber (in blue).

[Figure]

Figure C6. The concentrations of C1-C5 organic monoacid in the seeded (left panels) and nucleated (right panels) SOA experiments. The higher $NO_x$ input in E0314 (in blue) also led to higher ozone formation and eventually led to higher organic acids concentration resulting from ozonolysis reaction. Refer to Fig. C5 and text for more details.

*Overall, I would recommend the publication of the manuscript if the authors can address my questions and comments.*

Minor comments are show below.

*P3, line 89-91: H2O2 was introduced as a source of OH radical. Although OH exposure is presented in Table 1, H2O2 concentration itself is informative for readers.*

Reply:

The $H_2O_2$ concentration is not measured in the chamber. Based on the volume of injected $H_2O_2$ solution and the chamber volume, we estimated its concentration to be 30ppm. We have added in line91 that "The $H_2O_2$ concentration was roughly 30ppm estimated from the amount of its injection and the volume of chamber".

*P5, line 173, 179: It is curious why chloride ion was added in the estimation of NH4+, since there was no source of chloride in the smog chamber experiments.*

Reply:

There was a tiny amount of formed ammonium chloride after UV lights were switched on, e.g. in the nucleated SOA experiment (Fig. C7). Hence, we used it for ammonium estimation.

[Figure]

Figure C7. The concentration of chloride measured by AMS in the chamber.

*P6, Line 228: "Fig. 4" might a typo of "Fig. 6".*

Reply: We have fixed it.

*P7, line 270-274: Diacids such as malonic and succinic acids have vapor pressure similar to pinonic acid. Differently from pinonic acid, those diacids showed good correlation with ammonium. This might mean continuous formation of those diacids. It is not clear how these nonvolatile diacids could form in the gas phase until the later part of photooxidation.*
Reply:
We thank the referee for the insightful comment. The referee is correct to say that this indicates the continuous formation of those diacids in the gas phase at the later stage of photooxidation. Their formation mechanism is unclear for us and might be similar to the low MW monoacid. We rephrased the sentence in line 272-274 "The nice correlations of the gas-phase malonic and succinic acids to the particle-phase ammonium in this study suggest that diacids contribute to the formation of ammonium in both nucleated and seeded SOA experiments." to "The nice correlations of the gas-phase malonic and succinic acids to the particle-phase ammonium study suggest that diacids contribute to the formation of ammonium in both nucleated and seeded SOA experiments. It also indicates the continuous formation of those diacids at the later stage of photooxidation in nucleated and seeded experiments. The formation mechanism of these low molecule-weight diacids remains unclear for us and might be similar to the C1-C5 monoacids.".

*P9, Line 316: "affective" might be a typo of "effective".*
Reply: we have fixed it.

*P15, Table 1: a-Pinene might be a typo of "alpha-Pinene".*
Reply: we have fixed it.

*P18, Figure 3: In the caption, "SOA" needs to be deleted in ": : : (B) SOA seeded experiments : : :"*
Reply: we have fixed it.

*P20, Figure 5: In the caption, "the AMS NH4+ is the difference between NH4+,pre and NH4+,mea, refer to the text for details." needs to be clarified. In the text NH4+,pre and NH4+,mea are defined, whereas "AMS NH4+" is not defined. It should be defined clearly in the caption, e.g., NH4+,mea - NH4+,pre. It should be clarified in the caption of figures 6 and 7. It might be better to move the position of "x10-3" to improve the readability in the figure. It is same in figures 6 and 7.*
Reply:
Both referees brought up comments on the same question. Following the comment, we have renewed the X-axis with $NH_4^+{}_{\_orgacid}$ in Figs. 5-7 in both nucleated and seeded SOA experiments. $NH_4^+{}_{\_orgacid}$ is the amount of ammonium neutralized by the organic acids.

*P21, Figure 6: Readers might expect to see plots of AMS data also as Figure 5.*
Reply: we have fixed it.

*P22, Figure 7: Readers might expect to see plots of AMS data also as Figure 5.*
Reply: we have fixed it.

*It is worthy to clearly note in the text that all particulate data were not corrected to wall loss of particulate.*
Reply:
We have made it now clear in line 147, page 4 that "Additionally, the organics, nitrate and ammonium aerosols showed similar mass-based size distributions from our AMS measurement (Fig. 4), and internally-mixed aerosol can be assumed in our study, indicating that the aerosol components were lost in a similar rate to the chamber wall. Therefore, no loss correction to the aerosol wall disposition was conducted in this work."

*In the figure, axis titles need to be checked to properly note "NH4+", "SO42-", and "NO3-".*
Reply: we have fixed all of them.

Reference:
Friedman, B. and Farmer, D.K., SOA and gas phase organic acid yields from the sequential photooxidation of seven monoterpenes, Atmos. Environ., 187, 335-345, 2018.

Jacob, D. J. and Wofsy, S. C.: Photochemistry of biogenic emissions over the Amazon forest, J. Geophys. Res., 93, 1477–1486, doi:10.1029/JD093iD02p01477, 1988.

Kourtchev, I., Giorio, C., Manninen, A., Wilson, E., Mahon, B., Aalto, J., Kajos, M., Venables, D., Ruuskanen, T., Levula, J., Loponen, M., Connors, S., Harris, N., Zhao, D., Kiendler-Scharr, A., Mentel, T., Rudich, Y., Hullquist, M., Doussin J., Maenhaut, W., Back, J., Petaja, T., Wenger, J., Kulmala, M. and Kalberer, M., Enhanced volatile organic compounds emissions and organic aerosol mass increase the oliger content of atmospheric aerosols, Sci. Reports, 6, 35038, 2016.

Orzechowska, G.E., and Paulson, S.E.: Photochemical sources of organic acids. 1. Reaction of ozone with isoprene, propene, and 2-butenes under day and humid conditions using SPME, J. Phys. Chem. A, 109, 5358-5365, 2005.

Salonen, H., Pasanen, A., Lappalainen, S.K., Riuttala, H.M., Tuomi, T.M., Pasanen, P.O., Back, B.C. and Reijula, K.E., Airborne concentrations of volatile organic compounds, formaldehyde and ammonia in Finnish office buildings with suspected indoor air problems, Journal of Occupational and Environmental Hygiene, 6, 200-209, 2009.

Seinfeld, J.H. and Pandis, S.N., Atmospheric chemistryand physics: From air pollution to climate change, 3rd edition, John Wiley & Sons, Inc.,2016.

---

## Referee Report (RR1)

**comments on acp-2020-457-manuscript-version3**

In this study, Hao et al. studied the photochemical reaction of a-pinene in a smog chamber by using HR-ToF-PTRMS and HR-ToF-AMS. They mainly focused on the formation of ammonium. It was found the concentration of ammonium was in good correlation with gas phase organic acids. Thus, they pointed the important role of organic acids in the formation of ammonium. Considering the ubiquitous presence and its alkalinity in the atmosphere, this work provided useful information to realize the environmental impacts of NH3 and has important enlightenment for the future research on SOA formation. I think it is suitable for publication on ACP. However, there are still several concerns needed to be addressed before acceptance.

Major concerns.

1. the title. The "CCN-size" in the title is not necessary. The title of a paper should contain informative key words to represent the main content. However, no measurement about the CCN activity of particles was conducted. In fact, the size of particles reported in this study were based on the measurement limit of size-range of AMS, which has no direct link with CCN-size.

2. Since the work focused on the conversion of gas phase  $NH_3$  to particulate  $NH_4^+$ , the concentration of  $NH_3$  in the reaction system should be measured. In all experiments, ammonia was introduced to the chamber as an impurity. This condition makes the research loose and unscientific. The source of  $NH_3$  may include the evaporation from the wall of reactors due the previous deposition of ammonium. However, its sources may be not stable and fluctuate the concentration of  $NH_3$ . Then there are errors in the analysis of ammonium as a function of time due to the uncertainty of  $NH_3$ . At least the author needs to prove that the concentration of ammonia is constant or varies regularly during the experiment.

3. the interpretation about the difference between experimental results with and without seeds is lack. The authors designed two sets of experiments and found distinctly different results. However, the reasons were not provided. Why the VOC concentrations are so different between these two sets of experiments? What's the role and effects of seeds on the formation of ammonium? Moreover, it is difficult to understand the difference in the consumption of NOx ( $\Delta$ NOx-E314> $\Delta$ NOx-E327 in Fig. S2) and the corresponding formation of nitrate ( $\Delta$ NO3--E314< $\Delta$ NO3--E327 in Fig. 1).

4. removal Fig. 3 to SI. The data in fig. 3 were directly derived from the subtraction of data in Fig. 2, then no new meaning was provided in Fig. 3 and it could be removed to SI.

5. line 200: the statement "The delay is caused by the effect of nitric acid arising from the background NOx photooxidation." is not correct. According to Fig. 1-B2, it seems the main reason may be the formation of sulfate. However, no obvious consumption in

 $SO_2$  in Fig. S2 make the yield of sulfate hard to understand. Why the formation of ammonium in the seed experiments is earlier than sulfate, nitrate, and light on? Again, the effect of seeds needed to be explained.

After checking the authors' response to the former version, I find some critical concerns still present. A more convincing reply is needed to addressed these concerns.

Minor concerns:

- 1. line 9: ubiquitously
- 2. The Sequence of Fig. 3 appears in the main text is earlier than Fig. 4
- 3. line 168: Fig. 2?
- 4. line 247: AMSSurprisingly?

---

## Author Response (AR2)

**Responses to Referee 3:**

The authors thank the reviewer for her/his excellent comments.

*In this study, Hao et al. studied the photochemical reaction of a-pinene in a smog chamber by using HR-ToF-PTRMS and HR-ToF-AMS. They mainly focused on the formation of ammonium. It was found the concentration of ammonium was in good correlation with gas phase organic acids. Thus, they pointed the important role of organic acids in the formation of ammonium. Considering the ubiquitous presence and its alkalinity in the atmosphere, this work provided useful information to realize the environmental impacts of NH3 and has important enlightenment for the future research on SOA formation. I think it is suitable for publication on ACP. However, there are still several concerns needed to be addressed before acceptance.*

We thank the referee for the positive comment.

Major concerns.

*1.The title. The "CCN-size" in the title is not necessary. The title of a paper should contain informative key words to represent the main content. However, no measurement about the CCN activity of particles was conducted. In fact, the size of particles reported in this study were based on the measurement limit of size-range of AMS, which has no direct link with CCN-size.*

Reply:

We agree with the reviewer that the "CCN-size" in the title is not necessary. We have removed it and accordingly we modified the manuscript.

*2.Since the work focused on the conversion of gas phase NH3 to particulate NH4+, the concentration of NH3 in the reaction system should be measured. In all experiments, ammonia was introduced to the chamber as an impurity. This condition makes the research loose and unscientific. The source of NH3 may include the evaporation from the wall of reactors due the previous deposition of ammonium. However, its sources may be not stable and fluctuate the concentration of NH3. Then there are errors in the analysis of ammonium as a function of time due to the uncertainty of NH3. At least the author needs to prove that the concentration of ammonia is constant or varies regularly during the experiment.*

Reply:

The $NH_3$ was introduced to the chamber as an impurity. We lacked its measurement device in our lab and its concentration was estimated from AMS measurement. We agree with the referee that the missed $NH_3$ measurement could lead to some uncertainties of our interpretation to the $NH_4^+$ results. We need to point out that prior to each experiment, the chamber was continuously flushed overnight with laboratory clean air produced by a zero air generator to minimize any possible impurities from chamber air and wall. There is no physical reason for the fluctuation of the source of impurity: the temperature and other relevant conditions in the chamber were stable over the experiment. It is possible thought that the $NH_3$ concentration is gradually changing over the experiment, but this doesn't affect our conclusions. Fig C1A shows the time series of total organic acids measured by PTRMS (the sum of C1-C5 organic acids after excluding the background), and also the estimated mean $NH_3$ concentration of 19 ppb based on a statistics study on the ammonia in 14 Finnish office buildings (Salonen *et al.*, 2009). After 50 mins, the concentration of organic acids is higher than that of $NH_3$, and $NH_3$ is a limiting factor for the base-acid neutralization. Meanwhile, we still observe the formation of $NH_4^+$ in a continuous and smooth way in three experiments (Fig. C1B), suggesting a constant and regular source of $NH_3$.

[Figure]

[Figure]

Figure C1 (A)The time series of total C1-C5 organic acids after excluding the background measured by PTRMS in the nucleated SOA experiments, and the estimated mean concentration of $NH_3$ (in cyan) and the highest concentration (in orange), based on a statistics study on the ammonia in 14 Finnish office buildings in a national scale. (B) The measured ammonium concentration by AMS.

*3.The interpretation about the difference between experimental results with and without seeds is lack. The authors designed two sets of experiments and found distinctly different results. However, the reasons were not provided. Why the VOC concentrations are so different between these two sets of experiments? What's the role and effects of seeds on the formation of ammonium? Moreover, it is difficult to understand the difference in the consumption of NOx (ΔNOx-E314>ΔNOx-E327 in Fig. S2) and the corresponding formation of nitrate (ΔNO3--E314<ΔNO3--E327 in Fig. 1).*
Reply:
We thank the reviewer for the nice comments. In the seeded experiments αlpha-pinene concentration was about 20 times lower than in the nucleation experiment. The relatively low αlpha-pinene concentration was used to mimic an atmospherically relevant monoterpene mixing ratios, for example, in Hyytiälä forest area in Finland (e.g. Kourtchev *et al.*, 2006). We aimed to study if the ammonia plays a similar role as in the high a-pinene concentration case. Also, the nucleation experiments can't be conducted with such low concentrations because the nucleation experiments requires higher concentration due to the Kelvin effect to produce detectable amount of SOA mass in the size range of aerosol mass spectrometer (AMS). Both sets of experiments have consistently shown that organic acids were required to neutralize the observed ammonium salt and gas-phase organic acids contribute to the SOA formation.

[Figure]

Fig. C2. Time series of NOx and formed $O_3$ in E0314 and E0327.

Fig. C2 shows the added $NO_x$ and formed $O_3$ in the chamber in E0314 and E0327. The added NO was rapidly converted to $NO_2$ by $RO_2$ radicals derived from photooxidation of αlpha-pinene. The fate of $NO_2$ is a competition among the reactions (1)-(3):

$NO_2+h\nu \rightarrow NO+O$         (1a)
$O+O_2+M \rightarrow O_3+M$         (1b)
$NO_2+RO_2 \rightarrow$ N-containing organic compounds   (2a)
$NO+RO_2 \rightarrow$ N-containing organic compounds   (2b)
$NO_2+OH \rightarrow HNO_3$         (3a)
$HNO_3+NH_3 \rightarrow NH_4NO_3$      (3b)

In Fig. C2A, we have observed a large amount of $O_3$ formation because of reaction (1). Meanwhile, we also observed an increase in N:C ratios in both E0314 and E0327 either because of reaction (2) or $NH_3$ reaction as we have discussed in the manuscript. The results show the participation of NOx in different reactions, besides producing ammonium nitrate, which qualitatively explains the different amount of nitrate observed in two experiments (table C1).

[Figure]

Fig. C3. N:C ratios measured by AMS in E0314 and E0327.

Table C1. Changes of $NO_x$, and formed $O_3$ and nitrated aerosol in E0314 and E0327.

| Exp. | $\Delta NOx$, ppb | $\Delta O_3$, ppb | $NO_3$, µg m$^{-3}$ |
|------|-------------------|-------------------|----------------------|
| E0314 | 16 | 316 | 0.12 |
| E0327 | 7.3 | 52.3 | 2.5 |

Accordingly, we added in line89 in the manuscript that "The α-pinene concentration in the seeded experiments was about 20 times lower than in the nucleation experiment. The relatively low alpha-pinene concentration was used to mimic an atmospherically relevant monoterpene mixing ratios, for example, in Hyytiälä forest area in Finland (e.g. Kourtchev *et al.*, 2006). We aimed to study if the ammonia plays a similar role as in the high a-pinene concentration case."

*4. removal Fig. 3 to SI. The data in fig. 3 were directly derived from the subtraction of data in Fig. 2, then no new meaning was provided in Fig. 3 and it could be removed to SI.*
Reply:
The data in Fig. 3 was estimated from the conversion of the difference between predicted and measured ammonium in Fig. 2. The graph shows directly the amount of organic acids required to neutralize ammonium and the time at which the organic acids started to play a role in ammonium formation. Hence, we believe that the figure is informative and would like to keep it in the main text.

*5. line 200: the statement "The delay is caused by the effect of nitric acid arising from the background NOx photooxidation." is not correct. According to Fig. 1-B2, it seems the main reason may be the formation of sulfate. However, no obvious consumption in SO2 in Fig. S2 make the yield of sulfate hard to understand. Why the formation of ammonium in the seed experiments is earlier than sulfate, nitrate, and light on? Again, the effect of seeds needed to be explained.*
Reply:
As the referee pointed out that no obvious consumption in $SO_2$ in E0314-0316, the formation of sulfate was minor in the experiments. The increase in ammonium and sulphate concentrations before UV were switched on a in Fig 1-B2&A2 is because we added ammonium sulfate seeds to the chamber before the lights were turned on. We would like to note that the used ammonium sulphate seed and low organic concentrations makes the ammonium formation more difficult to distinguish from the data in the experiments E0314-E0316. Overall, the seed dominated the $NH_4^+$ signal in AMS. But e.g. in the case of E0316 we see the increase in $NH_4^+$ (after UV lights were turned on) and simultaneously a decrease in $SO_4^{2-}$ ($SO_4^{2-}$ signal is originated from the neutral ammonium sulphate seed) meaning that there is a $NH_4^+$ formation taking place in particle phase. Also at the same time the $NO_3^-$ and Org signals are increased suggesting that the increase in $NH_4^+$ is related to nitric acids and organic acids. The ammonium sulfate was added as seeds for condensational growth of SOA. Since the seeds were introduced to the chamber in a neutral state, we didn't observe other effects of seed aerosol, e.g., acid-catalysis accretion, on SOA formation.

[Figure]

Fig. C4. The time series of org, nitrate, sulfate and nitrate measured by AMS in the seeded SOA experiments.

After checking the authors' response to the former version, I find some critical concerns still present. A more convincing reply is needed to addressed these concerns.

Minor concerns:
1.line 9: ubiquitously
Reply: We have fixed it.

2. The Sequence of Fig. 3 appears in the main text is earlier than Fig. 4
Reply: We have fixed it.

3. line 168: Fig. 2?
Reply: We have fixed it.

4. line 247: AMSSurprisingly?
Reply: We have fixed it.

References:

[revised manuscript text omitted]

---

## Author Response (AR3)

**Responses to Referee 1:**

The authors thank the reviewer for her/his excellent comments. We have modified our manuscript according to the reviewer's comments. All the changes that we have made were saved as a format of "Track Changes" in the manuscript.

*This manuscript reports new findings on the role ammonia in the formation of SOA from photooxidation of alpha-pinene. Neutralization of carboxylic acid by ammonia in the gas phase is reported as a process of CCN-sized SOA formation. PTR-MS and AMS were mainly used to measure relevant species in gas and particle phase, respectively, and to interpret the measured data for the neutralization. Although this manuscript is unique and concise, there are several main questions to be clarified.*
*1) In this manuscript key data used to support the process are correlations between organic acids and particulate ammonium. Mixing ratio of ammonia might be a critical factor in the photooxidation of smog chamber. It is not clear about the introduction to a rector bag and mixing ratio of ammonia in the bag. The authors report N/C ratio remained consistent, implying negligible formation of organonitrogen via particle phase reaction. Reaction pathways might be largely dependent on the mixing ratio of ammonia.*

Reply:
Both referrers are concerned about the introduction of ammonia and its concentration in this study. We would like to point out that ammonia was not intentionally introduced to the chamber, but it was present as an impurity. Likely sources of ammonia were pressurized air, possible leakage of lab air, and water used for humidification of the chamber. We have added these in line92, page 3 that "In two sets of experiments, ammonia was introduced to the chamber as an impurity. The likely sources of ammonia were the pressurized air, possible leakage of the lab air and water used for humidification".

As we have stated in the manuscript in line 124-128 that we unfortunately lacked the measurement of $NH_3$ concentration in the chamber but estimated the minimum concentration of $NH_3$ present from our AMS measurement results, by assuming that the particulate ammonium salt ($NH_4^+$) was converted from the gas-phase $NH_3$ (Fig. 1). The maximum $NH_4^+$ concentration was in the range of 1.17~1.51 µg m$^{-3}$, which corresponds to a minimum $NH_3$ concentration level of 1.6 ~ 2.1 ppbV in our chamber. The $NH_3$ mixing ratio over continental range is typically between 0.1 and 10ppb (Seinfeld and Pandis, 2016). A statistics study on the ammonia concentration in 14 Finnish office buildings shows a range of 1-49 ug m$^{-3}$ and a geometric mean concentration of 14 ug m$^{-3}$ (corresponds 19 ppbV) (Salonen et al., 2009). Hence, our method should provide a lowest boundary of ammonia mixing ratio in the chamber. We have added the discussion in page 4 in the manuscript.

According to Salonen et al., the detected highest ammonia concentration at Finnish office building is 49 ug m$^{-3}$ (corresponding to 64ppb). This gives an approximatively higher limit for the order of expected ammonia concentration that we can use to perform the following analysis. Based on PTRMS measurements, we estimated the formed organic acid concentration to be about 50-250ppb at the end of our experiments. After comparison of the concentrations to the two species, the ammonia was a limiting species and would be neutralized by the organic acids at the end.

The referee's comments also guide us to recheck the N/C ration in both nucleated and seeded SOA experiments (Fig C1). Regardless of the relatively high noise level of our data, we can see the nearly constant or small time dependent increase in N/C ratio in both sets of experiments. Therefore, line 285 page 8, we have reworded the statement "After SOA are formed, carbonyl group of chemical compounds in SOA particles can also uptake $NH_3$ heterogeneously to form nitrogen-containing compounds (Zhu et al., 2018; Liu et al., 2015; Updyke et al., 2012; Dinar et al., 2007) and organic ammonium salts (Schlag et al., 2017). However, in this study the N:C ratios measured by AMS remained nearly constant at 0.002 for E0322 and E0326 and 0.004 for E0327 suggesting that the carbonyl-$NH_3$ heterogeneous reaction could be negligible." to "After SOA are formed, carbonyl group of chemical compounds in SOA particles can also uptake $NH_3$ heterogeneously to form nitrogen-containing compounds (Zhu et al., 2018; Liu et al., 2015; Updyke et al., 2012; Dinar et al., 2007) and organic ammonium salts (Schlag et al., 2017). Similarly, in this study the N:C ratios measured by AMS were nearly constant or slightly increasing in two sets of experiments (Fig. C1), suggesting the formation of nitrogen-containing compounds via carbonyl-$NH_3$ heterogeneous reaction.".

55       Definitely, we strongly agree with the referee that the accurate measurement of ammonia in the chamber is critical to understand the detailed mechanisms of neutralization of gas-phase organic acids, formation of particulate $CO_2^+$ and formation of NOC in the specific experiments.

[Figure]

60 Figure C1. The N:C ratios determined by AMS in (A) the nucleated SOA experiments, and (2) in the seeded SOA experiments.

*2) For seeded experiments alpha-pinene concentration was too low compared to nucleation experiments.*
65 *It is uncertain why such low concentration was used for seeded cases. It makes difficult to clearly show the temporal variation of organic acids. In seeded experiments correlation of organic acids with ammonium are not apparent as much as in nucleation experiments, probably due to lower concentration of alpha-pinene. A result stated in P5, line 181-183 (48.6 times lower concentration of CO2+ for seeded experiment) might be just due to lower initial α-pinene concentration for seeded condition (23 times lower).*
70 Reply:
We agree with the referee that in the seeded experiments αlpha-pinene concentration was about 20 times lower than in the nucleation experiment. The relatively low αlpha-pinene concentration was adopted to mimic an atmospherically relevant monoterpene mixing ratio, for example, in Hyytiälä forest area in Finland (e.g. Kourtchev *et al.*, 2006). We aimed to study if the ammonia plays a similar role as in the high
75 a-pinene concentration case. Meanwhile, in the presence of seed aerosol, the relative low a-pinene can produce detectable amount of secondary organic aerosol (SOA) for the aerosol mass spectrometer (AMS). As the referee pointed out, the correlation of organic acids with ammonium is less obvious in the seeded, low αlpha-pinene concentration experiments compared to the nucleation experiments. Part of reason is that in the seeded experiments, the produced ammonium resulting from organic acid neutralization ($NH_4^+{}_{orgacid}$)
80 is around 40 times less than in the nucleated cases. Such small amount of $NH_4^+{}_{orgacid}$ (at about 0.03 ug m$^{-3}$ level) made the determination to its concentration be noisy by AMS. Hence, its correlation to organic acids is worse in the seeded experiments than in the nucleated SOA experiments. Nevertheless, the results in two sets of experiments are consistent making our conclusion even stronger.

85 *3)*
*3.1) Figure 3 apparently show the difference in a lag time of CO2+_NH4. It was partly explained that nitric acid delayed its accumulation in high NOx condition. More explanation might be necessary for the delay in low NOx conditions.*
Reply:
90 A lag in time for $CO_2^+{}_{NH4}$ is also observed in the low $NO_x$ conditions, especially in the seeded experiments. The reasons for the observed delay is again affected by the nitric acid arising from the background $NO_x$ photooxidation, where we have observed particulate nitrate formation (Fig C2). The other reason is that the formation of particulate organic acids was much slower in the seeded experiments compared to the nucleated cases (Fig C3), implying that their reaction with ammonia was also delayed.
95     Accordingly, we have added the argument in line 189 " In the low-$NO_x$ test, a time lag for $CO_2^+{}_{NH4}$ is also observed, especially in the seeded SOA experiments. The delay is caused by the effect of nitric acid arising from the background $NO_x$ photooxidation. In addition, the slower formation of particulate organic

acids makes their reaction with ammonia delayed in the seeded experiments compared to the nucleated cases."

[Figure]

Figure C2. The formed nitrate aerosol measured by AMS in the seeded SOA experiment.

[Figure]

105      Figure C3. The measured $CO_2^+$ ion concentration measured by AMS (A) in the seeded and (B) in the nucleated SOA experiments. $CO_2^+$ ion was chosen to represent organic acids. The formation rate of $CO_2^+$ in seeded experiments was much slower than in the nucleated cases.

*3.2) Very limited amount of CO2+_NH4 was formed roughly at 1/100 of SOA in mass both in nucleation*
110      *and seeded experiments. Is it caused by limit in the available low MW carboxylic acids, ammonia, or other factors? It is probably associated with the contribution of neutralization and its atmospheric implication due to the omnipresence of NOx and nitric acids.*
Reply:
In the two sets of experiments, the total $CO_2^+$ ions account for 2.7% to 7.0% of SOA in mass (Fig. C4).
115      Out of the $CO_2^+$ ions, only $27.0 \pm 3.1$ % and $18.7 \pm 6.0$ % have participated in the reaction with ammonia in the nucleated and the seeded SOA experiments, respectively. This explains why $CO_2^+{}_{\_NH4}$ is roughly 1/100 of SOA in mass. In our experiments, the ammonia could be a limiting species. The produced low-MW carboxylic acid concentration is about one or two orders of magnitude greater than the high-MW acids. Hence, the reaction between low-MW organic acids and ammonia dominated over the reactions
120      between high-MW acids and ammonia. The vapor pressure of the formed low-MW organic ammonium salts is in the same order as their parental acids. Therefore, we speculate that a large amount of these low-MW organic ammonium salts were present in the gas phase and a minor amount of ammonium salts have participate on the particulate phase, contributing to $CO_2^+{}_{\_NH4}$. The amount of $CO_2^+{}_{\_NH4}$ is assumed to be related to: (1) the photooxidation and ozonolysis capacity in our chamber, which determines the amount
125      of particulate organic acids (total $CO_2^+$), (2) the relative amount of low- and high- MW organic acids and their physicochemical properties, (3) the relative amount of inorganic and organic acids, and (4) the relative amount of ammonia and total acids. We have added the discussion in line 190 in the manuscript.

[Figure]

Figure C4. The mass fraction of total $CO_2^+$ ions to the SOA mass.

*4) Although a previous paper (Friedman and Farmer, 2018) also reported the formation of low molecular weight organic acids as observed in this manuscript, little information is available about the formation mechanism of those species. Even a brief introduction of the formation mechanism would be very useful in the understanding of neutralization process.*

Reply:

The formation of low molecular weight organic acids remains unclear in the photochemical reaction of α-pinene (e.g.Friedman and Farmer, 2018). It can be speculated that these organic acids were produced as products of stabilized Criegee intermediate associated with ozonolysis of α-pinene and double carbon bond-containing products (Jacob and Wofsy, 1988; Orzechowska and Paulson, 2005). The statement is verified by the fact that we have observed the highest $O_3$ formation at the highest $NO_x/VOC_x$ initial input in E0314 (Figure C5). In E0314, we also have observed higher concentrations of C1-C5 monoacids corresponding to higher ozonolysis reactivity (Figure C6). The specific formation mechanisms of the organic acids need to be investigated in the future studies. We have included the discussion and the figures C5 and C6 in the manuscript.

[Figure]

Figure C5. The $O_3$ and $NO_x$ concentration in the chamber. The high $NO_x$ injection led to high $O_3$ concentration in the chamber (in blue).

[Figure]

Figure C6. The concentrations of C1-C5 organic monoacid in the seeded (left panels) and nucleated (right panels) SOA experiments. The higher $NO_x$ input in E0314 (in blue) also led to higher ozone formation and eventually led to higher organic acids concentration resulting from ozonolysis reaction. Refer to Fig. C5 and text for more details.

*Overall, I would recommend the publication of the manuscript if the authors can address my questions and comments.*

Minor comments are show below.
*P3, line 89-91: H2O2 was introduced as a source of OH radical. Although OH exposure is presented in Table 1, H2O2 concentration itself is informative for readers.*
Reply:
The $H_2O_2$ concentration is not measured in the chamber. Based on the volume of injected $H_2O_2$ solution and the chamber volume, we estimated its concentration to be 30ppm. We have added in line91 that "The $H_2O_2$ concentration was roughly 30ppm estimated from the amount of its injection and the volume of chamber".

*P5, line 173, 179: It is curious why chloride ion was added in the estimation of NH4+, since there was no source of chloride in the smog chamber experiments.*
Reply:
There was a tiny amount of formed ammonium chloride after UV lights were switched on, e.g. in the nucleated SOA experiment (Fig. C7). Hence, we used it for ammonium estimation.

[Figure]

175 Figure C7. The concentration of chloride measured by AMS in the chamber.

*P6, Line 228: "Fig. 4" might a typo of "Fig. 6".*

Reply: We have fixed it.
180

*P7, line 270-274: Diacids such as malonic and succinic acids have vapor pressure similar to pinonic acid. Differently from pinonic acid, those diacids showed good correlation with ammonium. This might mean continuous formation of those diacids. It is not clear how these nonvolatile diacids could form in the gas phase until the later part of photooxidation.*

185 Reply:
We thank the referee for the insightful comment. The referee is correct to say that this indicates the continuous formation of those diacids in the gas phase at the later stage of photooxidation. Their formation mechanism is unclear for us and might be similar to the low MW monoacid. We rephrased the sentence in line 272-274 "The nice correlations of the gas-phase malonic and succinic acids to the particle-phase
190 ammonium in this study suggest that diacids contribute to the formation of ammonium in both nucleated and seeded SOA experiments." to "The nice correlations of the gas-phase malonic and succinic acids to the particle-phase ammonium study suggest that diacids contribute to the formation of ammonium in both nucleated and seeded SOA experiments. It also indicates the continuous formation of those diacids at the later stage of photooxidation in nucleated and seeded experiments. The formation mechanism of these low
195 molecule-weight diacids remains unclear for us and might be similar to the C1-C5 monoacids.".

*P9, Line 316: "affective" might be a typo of "effective".*
Reply: we have fixed it.
200

*P15, Table 1: a-Pinene might be a typo of "alpha-Pinene".*
Reply: we have fixed it.

*P18, Figure 3: In the caption, "SOA" needs to be deleted in ": : : (B) SOA seeded experiments : : :"*
205 Reply: we have fixed it.

*P20, Figure 5: In the caption, "the AMS NH4+ is the difference between NH4+,pre and NH4+,mea, refer to the text for details." needs to be clarified. In the text NH4+,pre and NH4+,mea are defined, whereas "AMS NH4+" is not defined. It should be defined clearly in the caption, e.g., NH4+,mea - NH4+,pre. It*
210 *should be clarified in the caption of figures 6 and 7. It might be better to move the position of "x10-3" to improve the readability in the figure. It is same in figures 6 and 7.*
Reply:
Both referees brought up comments on the same question. Following the comment, we have renewed the X-axis with $NH_4^+$_orgacid in Figs. 5-7 in both nucleated and seeded SOA experiments. $NH_4^+$_orgacid is the
215 amount of ammonium neutralized by the organic acids.

*P21, Figure 6: Readers might expect to see plots of AMS data also as Figure 5.*
Reply: we have fixed it.

Reply: we have fixed it.

*It is worthy to clearly note in the text that all particulate data were not corrected to wall loss of particulate.*
Reply:
We have made it now clear in line 147, page 4 that "Additionally, the organics, nitrate and ammonium aerosols showed similar mass-based size distributions from our AMS measurement (Fig. 4), and internally-mixed aerosol can be assumed in our study, indicating that the aerosol components were lost in a similar rate to the chamber wall. Therefore, no loss correction to the aerosol wall disposition was conducted in this work."

*In the figure, axis titles need to be checked to properly note "NH4+", "SO42-", and "NO3-".*
Reply: we have fixed all of them.

Reference:

Friedman, B. and Farmer, D.K., SOA and gas phase organic acid yields from the sequential photooxidation of seven monoterpenes, Atmos. Environ., 187, 335-345, 2018.

Jacob, D. J. and Wofsy, S. C.: Photochemistry of biogenic emissions over the Amazon forest, J. Geophys. Res., 93, 1477–1486, doi:10.1029/JD093iD02p01477, 1988.

Kourtchev, I., Giorio, C., Manninen, A., Wilson, E., Mahon, B., Aalto, J., Kajos, M., Venables, D., Ruuskanen, T., Levula, J., Loponen, M., Connors, S., Harris, N., Zhao, D., Kiendler-Scharr, A., Mentel, T., Rudich, Y., Hullquist, M., Doussin J., Maenhaut, W., Back, J., Petaja, T., Wenger, J., Kulmala, M. and Kalberer, M., Enhanced volatile organic compounds emissions and organic aerosol mass increase the oliger content of atmospheric aerosols, Sci. Reports, 6, 35038, 2016.

Orzechowska, G.E., and Paulson, S.E.: Photochemical sources of organic acids. 1. Reaction of ozone with isoprene, propene, and 2-butenes under day and humid conditions using SPME, J. Phys. Chem. A, 109, 5358-5365, 2005.

Salonen, H., Pasanen, A., Lappalainen, S.K., Riuttala, H.M., Tuomi, T.M., Pasanen, P.O., Back, B.C. and Reijula, K.E., Airborne concentrations of volatile organic compounds, formaldehyde and ammonia in Finnish office buildings with suspected indoor air problems, Journal of Occupational and Environmental Hygiene, 6, 200-209, 2009.

Seinfeld, J.H. and Pandis, S.N., Atmospheric chemistryand physics: From air pollution to climate change, 3rd edition, John Wiley & Sons, Inc.,2016.

**Responses to Referee 2:**

265  The authors thank the reviewer for her/his excellent comments. We have modified our manuscript according to the reviewer's comments. All the changes that we have made were saved as a format of "Track Changes" in the manuscript.

270  *This manuscript reports chamber results of alpha-pinene SOA formation in the presence of ammonia, and suggested that organic acids have a central role in the formation of particle phase ammonium through neutralization. Major arguments supporting this conclusion come from the stoichiometric neutralization analysis, and correlations between gas-phase organic acids and particle-phase NH4+. The experiment looks carefully conducted and calibrated, with interesting results. However, the interpretation of the results is ambiguous and confusing. Although I trust in the data, I'm not convinced by the logic that draws*
275  *to the conclusions. The following concerns should be addressed before this work can be considered for publication for ACP.*
Reply:
We thank the referee for the positive comments.

280  Major concerns
*1. How did the authors estimate the NH3 concentration in the chamber "by assuming that the particulate ammonium salt (NH4+) was converted from the gas-phase NH3 (Fig. 1)"? Did the authors assume a 1:1 conversion ratio, namely all gas-phase NH3 are converted to NH4+? If so, this assumption is too arbitrary as NH3 is highly volatile. If the authors are introducing the NH3 from the "background gas", I'd suggest*
285  *at least an estimate of the typical NH3 level (e.g., through a supplementary measurement of typical background NH3 concentration in your lab, or provide the measured NH3 level in the vicinity environments), and do a calculation to check whether all the NH3 would have been depleted by the organic acids at your measured concentration level.*

290  Reply:
We would like to highlight that our estimation gives the lowest boundary for the possible $NH_3$ concentration. Hence, in our estimation we indeed assumed a conversion ratio of 1:1.
The ammonia was unintentionally introduced to the chamber quite likely with the pressurized air and water used for humidification. Unfortunately, we lacked the ammonia measurement device in our lab.
295  Based on a statistics study by Salonen et al (2009) on the ammonia in 14 Finnish office buildings where there were no specific emission source, the indoor concentrations varied between 1-49 ug m$^{-3}$ with a geometric mean concentration of 14 ug m$^{-3}$ (corresponds 19 ppbV). If we assume that ammonia concentrations in our experiments are on the same order with the highest value reported in Salonen et al. study, we can assume that in the beginning of photooxidation reaction after UV lights were switched on,
300  the ammonia concentration is higher than organic acid concentration, and organic acids should be the limiting species. As photooxidation reactions went on and organic acids formation accumulated in the chamber, their concentration was higher than ammonia, and the ammonia was a limiting species and would be neutralized by the organic acids at the end. However, it is still an open question about the distribution of organic ammonium: how much they are present in the gas phase and how much in the particle phase?
305  Accordingly, we have added in line these in line92, page 3 that "In two sets of experiments, ammonia was introduced to the chamber as an impurity. The likely sources of ammonia were the pressurized air, possible leakage of the lab air and water used for humidification".
And also in line 126, we added that "The maximum $NH_4^+$ concentration was in the range of 1.17~1.51 µg m$^{-3}$, which corresponds to a minimum $NH_3$ concentration level of 1.6 ~ 2.1 ppbV in our chamber. The
310  $NH_3$ mixing ratio over continental range is typically between 0.1 and 10 ppb (Seinfeld and Pandis, 2016). A statistics study on the ammonia concentration in 14 Finnish office buildings shows a range of 1-49 ug m$^{-3}$ and a geometric mean concentration of 14 ug m$^{-3}$ (corresponds 19 ppbV) (Salonen et al., 2009). Hence, our method should provide a lowest boundary of ammonia mixing ratio in the chamber."

315

*2. What on earth is the definition of CCN size in this paper?*
Reply:

The CCN size naturally depends on particle composition, but in general, it is often assumed that atmospheric aerosol particles larger than 100 nm are able to active to cloud droplets. AMS measurement
320 range is 35-1000 nm, and the aerosol mass is centered on the larger sizes, hence we can state that our measurements are in CCN size range. We have now clarified this in line 145, page 4 that "As the measured mass is centered on the larger sizes of AMS measurement range (35-1000nm, Jayne et al., 2000; Zhange et al., 2004), we can state that the measurements represents CCN relevant particle sizes (in general it is often assumed that atmospheric aerosol particles larger than 100 nm are able to active to cloud droplets)."
325

*3. The results seem to suggest that the measured NH4 concentration is more than enough to neutralize the inorganic acids (SO4, NO3 and Cl) (Eq. 1), while not enough to neutralize all organic acids ($CO_2^+$), as "the amount of $CO_2^+$ required for neutralizing ammonium accounted for the 27.0 _ 3.1 % of total $CO_2^+$ mass in the nucleated SOA experiments, and 18.7 _ 6.0 % in the seeded SOA experiments". If so, then*
330 *ammonia would be the limiting species, the concentration of which should be determined by available total ammonia concentrations. In this case, it is confusing why the $NH_4^+$ would correlated to gas-phase organic acids. If the authors don't really think the CO2+ all comes from organic acids, it should be clarified in the manuscript.*
Reply:
335 We thank the referee for the excellent comment and the next one. The referee is correct that $NH_4^+$ is sufficient to neutralize the inorganic acids, but not adequate to neutralize organic acids, and it is a limiting species, as we have speculated in the reply to the first general comment.

We would first make it clear that in our study the $NH_4^+$ is observed to correlate to the C1-C5 monoacids and two diacids, but not all organic acids, e.g. no correlation to pinonic acid. Fig. C1 shows time series of
340 the particle-phase total $CO_2^+{}_{\_total}$ (green), the ammonium resulting from organic acids ($NH_4^+{}_{\_orgacid}$) , and gas-phase formic acid and pinonic acid in the nucleated SOA experiments. The formic acids was selected to represent the low molecule-weight (MW) organic monoacids, and pinonic acid (molecule weight is 184) was chosen to represent high MW organic acid. We can see that:

(1) The time series of the total particulate organic acids ($CO_2^+{}_{\_total}$ in Fig. C1) is not following the low
345 MW organic acids (formic acid), nor the high MW organic acids (pinonic acid), suggesting that particulate organic acids were not derived from one single type of gas organic acids.

(2) The formation of $NH_4^+{}_{\_orgacid}$ is mainly related to the low MW organic monoacids. We can see that the trend of formic acid formation is consistent with $NH_4^+{}_{\_orgacid}$. In the experiments, we observed a high production of these low MW organic monoacids: their concentrations were about one or two
350 orders of magnitude greater than the concentration of high MW acids. Hence, compared with the high MW acids, the low MW monoacids dominated the reaction with ammonia in the gas phase if we assume similar reaction rate constants $k_{orgacid}$ for both low- and high- MW acids with ammonia ($R=k_{orgacid}\cdot[OrgAcid]\cdot[NH_3]$), and the majority of $NH_4^+{}_{\_orgacid}$ are associated with the low MW monoacids. Therefore, $NH_4^+{}_{\_orgacid}$ is well correlated to the low-MW organic monoacids. We need
355 to point out that although the low MW organic monoacids dominate the ammonia reaction, the formed ammonium salts are still volatile, so we speculate that a large amount of them are present in the gas phase.

(3) Inconsistency between the formation of high MW organic acids and the $NH_4^+{}_{\_orgacid}$. Here we use pinonic acid as an example because it is a high MW organic acid that has been measured in our
360 work. The vapor pressure of pinonic acid is about 4-5 orders of magnitude lower than those of C1-C5 monoacids (Jimenez et al., 2009), and its reactivity to ammonia is more than 500 times lower than that of C1-C5 monoacids. This allows pinonic acid to condense directly on the particle phase before it reacts in large amount with ammonia. These condensed high-MW organic acids are efficient to fragment to form $CO_2^+$ in AMS, and produced more than 73.0 % and 81.3% of the
365 observed $CO_2^+$ in the nucleated and seeded SOA experiments, respectively. The remaining $CO_2^+$ signals (less than 27.0 % and 18.7 %) came from the interaction between low MW organic acids and ammonia. Our analysis is also consistent with other studies showing that the $CO_2^+$ detected in AMS is caused by the thermal decomposition of the mono-, di-, and poly carboxylic acid groups, and is related to high MW oxygenated organic species (e.g. Zhang et al., 2005; Alfarra et al., 2004).
370 We still want to point out that the $CO_2^+{}_{\_total}$ accounts for 2.7% to 7.0% of SOA mass at maximum in each experiments, and it is very possible that the high-MW organic acids contributed to the majority of these $CO_2^+{}_{\_total}$.

So, our above analysis interprets the observation that $NH_4^+{}_{\_orgacid}$ correlates well to the low MW organic monoacids, even $CO_2^+{}_{\_NH4}$ accounts for only 27.0 % and 18.7 % of the total $CO_2^+$.

[Figure]

Figure C1. The evolution of time series of the particle-phase total $CO_2^+{}_{\_total}$ (green), the formed ammonium attributed to organic acid neutralization ($NH_4^+{}_{\_orgacid}$), and gas-phase formic acid and pinonic acid in the nucleated SOA experiments. Formic acid is chosen to represent C1-C5 organic monoacids and pinonic acid as a high molecule-weight organic acid.

*4. Following above question, the good correlations among gas-phase organic acids and NH4+ should be expected if the gas-phase organic acids are the limiting species, with total ammonium more than enough to influence the partitioning equilibriums. The results seem self-conflicting.*
Reply:
As we have replied to the prior comment, ammonia is the limiting species in our experiments. The majority of $NH_4^+{}_{\_orgacid}$ are associated with the low MW monoacids, and the high-MW organic acids contributed to the majority of the observed $CO_2^+$. Our analysis is consistent with the observation that $NH_4^+{}_{\_orgacid}$ correlates well to the low MW organic monoacids.

*5. The authors argue that "The reaction of HNO3 and NH3 takes precedence over the reaction between organic acids and NH3" (Line 186). Fig. 1 also shows formation of sulfate and nitrate from photooxidation of SO2 and NOx. If so, the influences from inorganics should be first excluded in analyzing the correlations of NH4+ and gas phase organic acids (Fig. 5). Instead of AMS NH4+, the difference between measuredNH4+ and that predicted by stoichiometric neutralization analysis of inorganic acid (Eq.1) should be used (i.e., Free NH4+ = NH4+,meas - NH4+,pre). This is similar concept with the correction of (NH4)2SO4 for the seeded experiment in Fig. 5. How would the correlation look like after this kind of correlation?*
Reply:
Following the referee's comment, in the nucleated SOA experiments, the sulfate, nitrate and chloride related ammonium was excluded from the $NH_4^+{}_{\_mea}$, so only organic acid-related ammonium was used for plotting in Figs. 5-7. The identical analysis protocol was applied in both nucleated and seeded SOA experiments. After we did the analysis suggested by the referee, the correlation coefficients don't change much (table C1). Meanwhile, we also noticed an error in the seeded SOA experiments, that is, we misused ammonium-related organic acids for X-axis in Figs. 5-7; Instead, the correct one should be organic acidsrelated ammonium. The mistake doesn't affect any of our conclusion, but after correction, the correlation relationship is becoming better in the figures.

410

Table C1. The correlation coefficients in the new and old figures.

| Correlation species | Correlation coefficient ($r^2$) | | | |
|---|---|---|---|---|
| | Nucleated SOA(old) | Nucleated SOA(new) | Seeded SOA(old) | Seeded SOA(new) |
| Butyric acid vs $NH_4^+{}_{orgacid}$ (Fig.5) | 0.97-0.97 | 0.88-0.95 | 0.68-0.73 | 0.76-0.83 |
| $C_4H_8O_2^+$ vs $NH_4^+{}_{orgacid}$ (Fig.5) | 0.98-0.99 | 0.90-0.97 | 0.04-0.74 | 0.12-0.80 |
| Pentanoic acid vs $NH_4^+{}_{orgacid}$ (Fig.5) | 0.89-0.94 | 0.84-0.92 | 0.22-0.63 | 0.27-0.62 |
| $C_5H_{10}O_2^+$ vs $NH_4^+{}_{orgacid}$ (Fig.5) | 0.90-0.94 | 0.84-0.91 | 0.05-0.10 | 0.0-0.14 |
| Formic acid vs $NH_4^+{}_{orgacid}$ (Fig.6) | 0.94-0.99 | 0.95-0.97 | 0.76-0.86 | 0.81-0.87 |
| Acetic acid vs vs $NH_4^+{}_{orgacid}$ (Fig.6) | 0.89-0.98 | 0.97-0.98 | 0.74-0.84 | 0.80-0.84 |
| Propionic acis vs $NH_4^+{}_{orgacid}$ (Fig.6) | 0.91-0.98 | 0.96-0.97 | 0.69-0.74 | 0.72-0.79 |
| Succinic acid vs $NH_4^+{}_{orgacid}$ (Fig.7) | 0.94-0.97 | 0.91-0.94 | 0.70-0.81 | 0.51-0.81 |
| Malonic acid vs $NH_4^+{}_{orgacid}$ (Fig.7) | 0.95-0.96 | 0.79-0.90 | 0.44-0.68 | 0.42-0.67 |

Minor concerns

415 *1.There're some typos in the manuscript. For example, Line 59, "updake" should be "uptake". Line 131, "aftert" should be "after". Line 135, there's duplicate periods. The manuscript should be read through more carefully.*

Reply:

The typos were corrected and we have read the manuscript carefully.

420

*2. I'd suggest name the predicted NH4+ differently for that based on Eq. 1 and Eq. 2, i.e. that predicted with / without consideration of CO2+.*

Reply:

We have changed $NH_4^+{}_{,pre}$ to $NH_4^+{}_{,pre\_CO2}$ in Eq. 2.

425

Reply:

The $NH_3$ was introduced to the chamber as an impurity. We lacked its measurement device in our lab and its concentration was estimated from AMS measurement. We agree with the referee that the missed $NH_3$ measurement could lead to some uncertainties of our interpretation to the $NH_4^+$ results. We need to point out that prior to each experiment, the chamber was continuously flushed overnight with laboratory clean air produced by a zero air generator to minimize any possible impurities from chamber air and wall. There is no physical reason for the fluctuation of the source of impurity: the temperature and other relevant conditions in the chamber were stable over the experiment. It is possible thought that the $NH_3$ concentration is gradually changing over the experiment, but this doesn't affect our conclusions. Fig C1A shows the time series of total organic acids measured by PTRMS (the sum of C1-C5 organic acids after excluding the background), and also the estimated mean $NH_3$ concentration of 19 ppb based on a statistics study on the ammonia in 14 Finnish office buildings (Salonen *et al.*, 2009). After 50 mins, the concentration of organic acids is higher than that of $NH_3$, and $NH_3$ is a limiting factor for the base-acid neutralization. Meanwhile, we still observe the formation of $NH_4^+$ in a continuous and smooth way in three experiments (Fig. C1B), suggesting a constant and regular source of $NH_3$.

[Figure]

[Figure]

Figure C1 (A)The time series of total C1-C5 organic acids after excluding the background measured by PTRMS in the nucleated SOA experiments, and the estimated mean concentration of $NH_3$ (in cyan) and the highest concentration (in orange), based on a statistics study on the ammonia in 14 Finnish office buildings in a national scale. (B) The measured ammonium concentration by AMS.

*3.The interpretation about the difference between experimental results with and without seeds is lack. The authors designed two sets of experiments and found distinctly different results. However, the reasons were not provided. Why the VOC concentrations are so different between these two sets of experiments? What's the role and effects of seeds on the formation of ammonium? Moreover, it is difficult to understand the difference in the consumption of NOx ($\Delta$NOx-E314>$\Delta$NOx-E327 in Fig. S2) and the corresponding formation of nitrate ($\Delta$NO3--E314<$\Delta$NO3--E327 in Fig. 1).*

Reply:

We thank the reviewer for the nice comments. In the seeded experiments αlpha-pinene concentration was about 20 times lower than in the nucleation experiment. The relatively low αlpha-pinene concentration was used to mimic an atmospherically relevant monoterpene mixing ratios, for example, in Hyytiälä forest area in Finland (e.g. Kourtchev *et al.*, 2006). We aimed to study if the ammonia plays a similar role as in the high a-pinene concentration case. Also, the nucleation experiments can't be conducted with such low concentrations because the nucleation experiments requires higher concentration due to the Kelvin effect to produce detectable amount of SOA mass in the size range of aerosol mass spectrometer (AMS). Both sets of experiments have consistently shown that organic acids were required to neutralize the observed ammonium salt and gas-phase organic acids contribute to the SOA formation.

[Figure]

Fig. C2. Time series of NOx and formed $O_3$ in E0314 and E0327.

Fig. C2 shows the added $NO_x$ and formed $O_3$ in the chamber in E0314 and E0327. The added NO was rapidly converted to $NO_2$ by $RO_2$ radicals derived from photooxidation of αlpha-pinene. The fate of $NO_2$ is a competition among the reactions (1)-(3):

$$NO_2+hv \rightarrow NO+O \qquad (1a)$$
$$O+O_2+M \rightarrow O_3+M \qquad (1b)$$
$$NO_2+RO_2 \rightarrow \text{N-containing organic compounds} \quad (2a)$$
$$NO+RO_2 \rightarrow \text{N-containing organic compounds} \quad (2b)$$
$$NO_2+OH \rightarrow HNO_3 \qquad (3a)$$
$$HNO_3+NH_3 \rightarrow NH_4NO_3 \qquad (3b)$$

In Fig. C2A, we have observed a large amount of $O_3$ formation because of reaction (1). Meanwhile, we also observed an increase in N:C ratios in both E0314 and E0327 either because of reaction (2) or $NH_3$ reaction as we have discussed in the manuscript. The results show the participation of NOx in different reactions, besides producing ammonium nitrate, which qualitatively explains the different amount of nitrate observed in two experiments (table C1).

[Figure]

Fig. C3. N:C ratios measured by AMS in E0314 and E0327.

Table C1. Changes of $NO_x$, and formed $O_3$ and nitrated aerosol in E0314 and E0327.

| Exp. | $\Delta NOx$, ppb | $\Delta O_3$, ppb | $NO_3$, $\mu g\ m^{-3}$ |
|------|------|------|------|
| E0314 | 16 | 316 | 0.12 |
| E0327 | 7.3 | 52.3 | 2.5 |

540

Accordingly, we added in line89 in the manuscript that "The α-pinene concentration in the seeded experiments was about 20 times lower than in the nucleation experiment. The relatively low alpha-pinene concentration was used to mimic an atmospherically relevant monoterpene mixing ratios, for example, in Hyytiälä forest area in Finland (e.g. Kourtchev *et al.*, 2006). We aimed to study if the
545    ammonia plays a similar role as in the high a-pinene concentration case."

*4. removal Fig. 3 to SI. The data in fig. 3 were directly derived from the subtraction of data in Fig. 2, then no new meaning was provided in Fig. 3 and it could be removed to SI.*
Reply:
The data in Fig. 3 was estimated from the conversion of the difference between predicted and measured
550    ammonium in Fig. 2. The graph shows directly the amount of organic acids required to neutralize ammonium and the time at which the organic acids started to play a role in ammonium formation. Hence, we believe that the figure is informative and would like to keep it in the main text.

*5. line 200: the statement "The delay is caused by the effect of nitric acid arising from the background*
555    *NOx photooxidation." is not correct. According to Fig. 1-B2, it seems the main reason may be the formation of sulfate. However, no obvious consumption in SO2 in Fig. S2 make the yield of sulfate hard to understand. Why the formation of ammonium in the seed experiments is earlier than sulfate, nitrate, and light on? Again, the effect of seeds needed to be explained.*
Reply:
560    As the referee pointed out that no obvious consumption in $SO_2$ in E0314-0316, the formation of sulfate was minor in the experiments. The increase in  ammonium and sulphate concentrations before UV were switched on a in Fig 1-B2&A2 is because we added ammonium sulfate seeds to the chamber before the lights were turned on.  We would like to note that the used ammonium sulphate seed and low organic concentrations makes the ammonium formation more difficult to distinguish from the data in the
565    experiments E0314-E0316. Overall, the seed dominated the $NH_4^+$ signal in AMS. But e.g. in the case of E0316 we see the increase in $NH_4^+$ (after UV lights were turned on) and simultaneously a decrease in $SO_4^{2-}$ ($SO_4^{2-}$ signal is originated from the neutral ammonium sulphate seed) meaning that there is a $NH_4^+$ formation taking place in particle phase. Also at the same time the $NO_3^-$ and Org signals are increased suggesting that the increase in $NH_4^+$ is related to nitric acids and organic acids. The ammonium sulfate
570    was added as seeds for condensational growth of SOA. Since the seeds were introduced to the chamber in a neutral state, we didn't observe other effects of seed aerosol, e.g., acid-catalysis accretion, on SOA formation.

[Figure]

Fig. C4. The time series of org, nitrate, sulfate and nitrate measured by AMS in the seeded SOA experiments.

After checking the authors' response to the former version, I find some critical concerns still present. A more convincing reply is needed to addressed these concerns.

Minor concerns:
1.line 9: ubiquitously
Reply: We have fixed it.

2. The Sequence of Fig. 3 appears in the main text is earlier than Fig. 4
Reply: We have fixed it.

3. line 168: Fig. 2?
Reply: We have fixed it.

4. line 247: AMSSurprisingly?
Reply: We have fixed it.

References:

[revised manuscript text omitted]